# Stress-induced phase separation in plastics drives the release of amorphous polymer micropollutants into water

Dunzhu Li [1,2,3] ✉, Peijing Li[4], Yunhong Shi[3], Emmet D. Sheerin [2,5], Zihan Zhang[3], Luming Yang[2,3], Liwen Xiao [3,6] ✉, Christopher Hill[2,5], Conall Gordon[2,5], Manuel Ruether[5], Joshua Pepper[2,5], John E. Sader [7], Michael A. Morris [2,5], Jing Jing Wang [2] ✉ & John J. Boland [2,5] ✉

Residual stress is an intrinsic property of semicrystalline plastics such as polypropylene and polyethylene. However, there is no fundamental understanding of the role intrinsic residual stress plays in the generation of plastic pollutants that threaten the environment and human health. Here, we show that the processing-induced compressive residual stress typically found in polypropylene and polyethylene plastics forces internal nano and microscale segregation of low molecular weight (MW) amorphous polymer droplets onto the plastic's surface. Squeeze flow simulations reveal this stress-driven volumetric flow is consistent with that of a Bingham plastic material, with a temperature-dependent threshold yield stress. We confirm that flow is thermally activated and stress dependent, with a reduced energy barrier at higher compressive stresses. Transfer of surface segregated droplets into water generates amorphous polymer micropollutants (APMPs) that are denatured, with structure and composition different from that of traditional polycrystalline microplastics. Studies with water-containing plastic bottles show that the highly compressed bottle neck and mouth regions are predominantly responsible for the release of APMPs. Our findings reveal a stress-induced mechanism of plastic degradation and underscore the need to modify current plastic processing technologies to reduce residual stress levels and suppress phase separation of low MW APMPs in plastics.

Semicrystalline plastics, e.g., polypropylene (PP) and polyethylene (PE), are pervasively used in modern water systems for various applications, including drinking water supply (PE and PP pipes), water storage (PP bottles, containers, and caps), heating systems (cross-linked PE-plumbing pipes) and wastewater discharge[1–4]. Despite the advantages of plastics in these applications, degradation can result in the formation of plastic pollutants that escape into the environment. The release of pollutants, such as microplastics (MPs), nanoplastics, and plastic additives whenever water comes into contact with plastics, is a significant public concern[5,6]. The ubiquitous presence of plastic pollutants in the environment (of which 71% in the ocean are PP or PE[7]) has led to their penetration into global food chains and emerged as a

[1]Jiyang College, Zhejiang A&F University, Zhuji, China. [2]AMBER Research Centre and Centre for Research on Adaptive Nanostructures and Nanodevices (CRANN), Trinity College Dublin, Dublin, Ireland. [3]Department of Civil, Structural and Environmental Engineering, Trinity College Dublin, Dublin, Ireland. [4]School of Mathematics and Statistics, The University of Melbourne, Victoria, Australia. [5]School of Chemistry, Trinity College Dublin, Dublin, Ireland. [6]TrinityHaus, Trinity College Dublin, Dublin, Ireland. [7]Graduate Aerospace Laboratories and Department of Applied Physics, California Institute of Technology, Pasadena, USA. ✉e-mail: lidunzhu@zafu.edu.cn; liwen.xiao@tcd.ie; jjwang@tcd.ie; jboland@tcd.ie

threat to the environment, wildlife, and human health[5,8,9]. The degradation and leaching of plastic waste and the wear and breakdown of plastic products and clothing are important sources of plastic pollutants[10,11], including the use of plastic products for drinking water supply (pipes[1]), storage (bottles[4,12], cups[13], and containers[14]) and heating (kettles[15,16]) that can lead to direct MP ingestion by humans[12,15,17–19]. Despite the myriads of potential sources, most studies have focused on degradation mechanisms induced by extrinsic factors, such as UV irradiation and/or mechanical abrasion[20–22]. Currently, there is no fundamental understanding of how intrinsic factors such as residual stress influence the generation of plastic pollutants.

Modern plastics are a complex cocktail of polymers, chemical additives, and residual monomers. Polymer constituents are typically semicrystalline (crystallinity range: 20–80%) and comprised of a spherulitic arrangement of alternating (nanoscale thick) crystalline and amorphous layers[23–25]. Fabrication of non-fiber-based plastics involves cooling molten resins (in processes such as extrusion) containing blended polymers and additives into desired shapes, which introduces the residual stresses that are pervasively found in plastic products[26,27]. Residual stress refers to the locked-in stress distribution present in a structure, component, plate or sheet, while there are no external loads or forces applied[27]. The generation of high levels of residual surface stress is the result of multiple factors such as volumetric shrinkage, flow-induced stresses, crystallization, and heat transfer[28,29]. For example, differential cooling and heat transfer cause the surface region of the resin to cool more quickly and shrink relative to its interior, resulting in compressive stresses at the surfaces of plastic parts, typically 10 to 30 MPa[29]. Even higher levels of residual stress are generated whenever plastic parts are engineered into particular shapes, such as the neck and mouth regions of plastic bottles[30].

To unravel the impact of intrinsic residual stress on the release of plastic pollutants, we conducted cantilever beam experiments and a squeeze flow numerical study that show residual stress drives nano and microscale phase separation of low-molecular weight (MW) amorphous polymer onto the surfaces of plastics, from where it can be released as an emergent kind of amorphous polymer micropollutant (APMP). We characterized the viscoplastic flow behaviours at the surfaces of standard PP and PE, which together account for 54% of global plastics production[31]. We measured the rate of stress-driven flow onto the surface, including the activation energy for flow and showed that the amorphous polymer behaves like a Bingham plastic material[32], characterized by a temperature-dependent threshold stress. Crucially, we demonstrate that stress-driven phase separation is an important generator of denatured APMPs, whenever plastics come into contact with water.

## Results

### Segregation of the amorphous polymer to plastic surfaces

We began with pieces of standard semicrystalline PP sheets (Goodfellow, 40 mm × 3 mm, 0.5 mm thickness, crystallinity 49.7%, Suppl. Fig. 1) that were formed into cantilevers by clamping one end while vertically deflecting the other end (the free end, to a height of 20 mm, Fig. 1a, Suppl. Fig. 2, Suppl. Table 1). The stressed PP cantilevers were then separately placed in ovens (under air ambient) at temperatures ranging from 35 to 95 °C, typical of that experienced during a plastic's lifecycle (i.e., transport, storage, daily use food preparation and discarded in the environment). After 1 h at 95 °C, the upper compressive side of the cantilever close to the clamped end (-5 mm from the clamped end) developed a significant quantity of droplet-like features with typical lateral sizes of 0.5–1 μm and heights of 40–70 nm (Fig. 1b–g). After increasing the exposure time to 4 h, the droplets grew to have lateral sizes of 1.5 μm and heights of 90–120 nm. Interestingly, no droplets were found across the lower tensile surface of the cantilevered PP sheet, regardless of exposure time. Similar experiments performed for different time durations at temperatures of 85, 60 and 35 °C (Suppl. Fig. 3)

confirmed that the quantity and size of the emergent droplets increased with temperature, consistent with an activated stress-driven flow of droplets on the compressed side of the PP sheets. To comprehensively investigate the impact of various factors on the emergence of droplets, experiments were designed and performed using a full factorial methodology with four factors (Suppl. Fig. 4, Suppl. Table 2). ANOVA analysis reveals that phase separation and droplet formation are not dependent on the ambient environment but are controlled by temperature and the local stress condition at the surface (see below).

Rheological measurements showed that these droplets are sticky gel-like substances that can be easily removed from the PP sheet and transferred to an aluminum foil, which is an excellent substrate for optical characterization. Raman spectroscopy analysis showed the droplets closely matched the spectrum of standard amorphous PP wax (Goodfellow MW ≈ 10,000 g mol⁻¹, Suppl. Fig. 1), with a hit quality index (HQI) of 0.93 in the low wavenumber range (800–1500 cm⁻¹, Fig. 1h). Crucially, there are clear differences between spectra recorded from these droplets and the parent PP cantilever beam, with substantial differences at 810 and 840 cm⁻¹ associated with the degree of PP crystallinity and the absence of peak features at 1168 cm⁻¹ associated with the crystalline C-C backbone vibration[33,34] (see highlighted regions in Fig. 1h). A detailed Raman and FTIR spectra analysis in Suppl. Figs. 5, 6 and Suppl. Note 1 confirmed that these droplets are primarily composed of amorphous PP (a-PP). Ethanol soaking, previously shown to dissolve small molecular chemical additives[31,35], resulted in no observable change in the sizes of droplets generated at temperatures between 35–85 °C. Droplets generated at 95 °C showed a modest volume reduction (typically ≈7.3%) after 5-h ethanol exposure (Suppl. Fig. 7).

Gel permeation chromatography (GPC) was used to determine the MW and molecular weight distribution (MWD) of the original PP plastic sheet and the separated droplets. The sheet has an average MW of 97,000 g mol⁻¹ and a dispersity (Đ) of 3.6 (see Suppl. Fig. 8), consistent with a wide distribution of polymer chain lengths. The droplet GPC curve showed a main peak and a smaller peak, indicating two distinct MW distributions (Fig. 1i). The main peak, which accounted for 79% of the total mass of the droplets, corresponded to an average MW of 22,000 g mol⁻¹ (degree of polymerization $n = 522$), which is significantly less than the average MW of the original sheet. The minor peak had an average MW of 510 g mol⁻¹ (Suppl. Note 2). The main GPC peak is evidently due to the presence of low MW polymers in the origin PP sheet that either comprised the amorphous matrix or were generated under flow conditions by shearing off weakly bound short polymer chains from crystalline domains due to their reduced van der Waals interactions[36]. The minor peak is likely due to chemical additives that are insoluble in alcohol[37] or possibly oligomers with $n \approx 12$. We note that oligomer release has been reported for plastics in contact with oils and fats during microwave exposure[37]. These data unequivocally demonstrate the significant mobility of short-chain polymers and molecular species in plastics under modest stress conditions and the denatured state of the plastic-derived droplets at the surface.

### Analysis of stress-driven amorphous phase separation

The stress along a cantilever is known[38–40] to vary linearly from a maximum value at the clamped end to zero stress at the free end. To quantify the influence of this stress on the cantilever's compressive surface, we measured the spatial distribution of a-PP droplets along the cantilever. The number, size and shape of the droplets were determined at each location, and their volume was estimated using the ellipsoidal method (Suppl. Fig. 9, ethanol treatment was used to remove any chemical additives present). The results are shown in Fig. 2a–h for a cantilever that was heated for 4 h at 95 °C. The spatial distribution in Fig. 2a–g shows the greatest density of droplets close to the clamped end and that the number and size of the droplets continuously decreased along the cantilever up to -5 mm from the free end, beyond which there was no marked phase separation. Time-

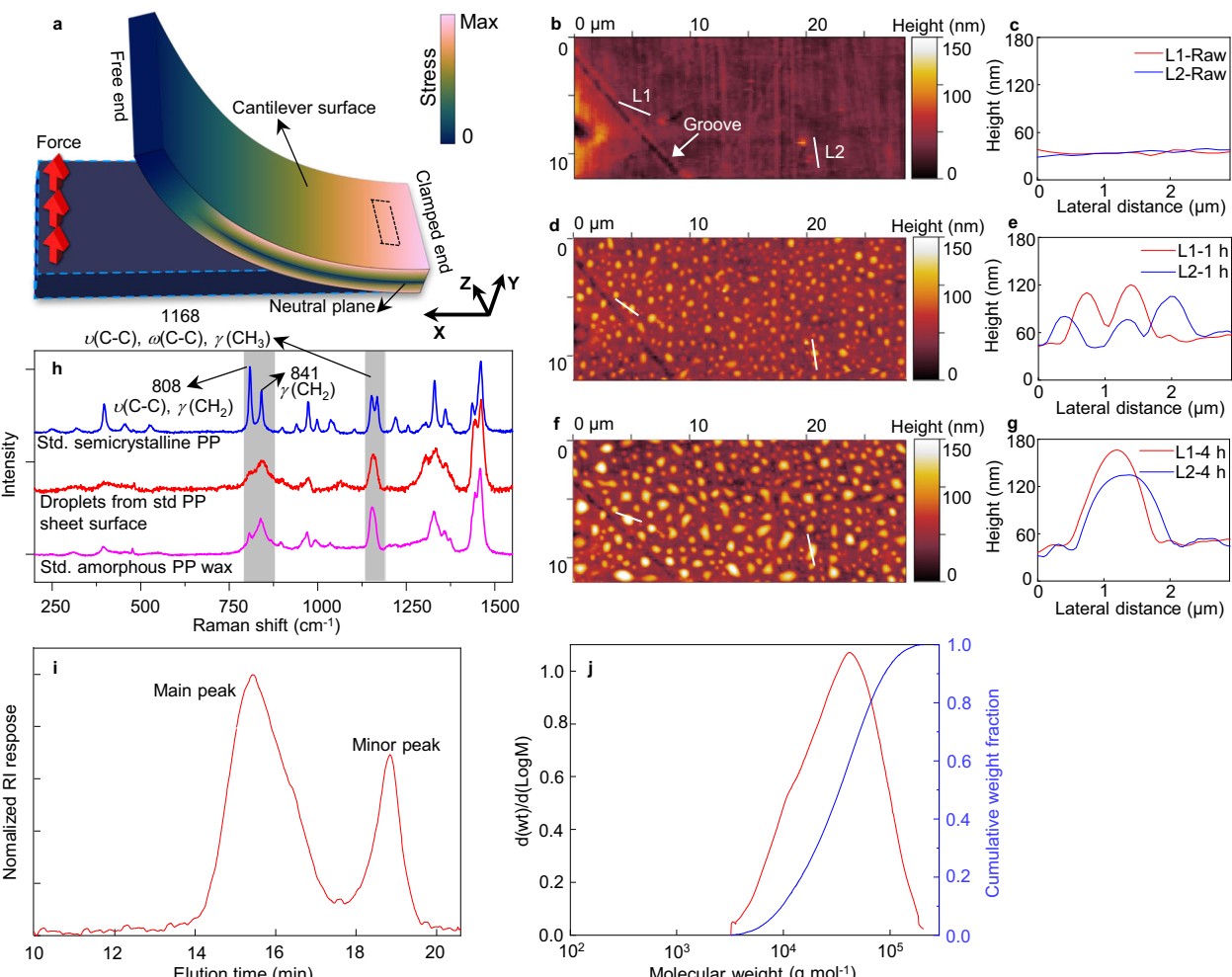

**Fig. 1 | Surface stress drives amorphous polypropylene phase separation from semicrystalline polypropylene at 95 °C. a** Schematic of the cantilever test configuration and the associated stress distribution. Black box on cantilever surface indicates the location of the AFM test in Fig. 1b–g. **b**–**g** AFM in-situ tests at the same location (see arrow marker) on the compressive side of the cantilever around 5 mm from the clamped end; **b**, **c** before, **d**, **e** after 1 h and **f**, **g** 4 h in a 95 °C oven, respectively. **c**, **e**, **g** The red and blue lines correspond to L1 and L2, respectively, as indicated in **b**. **h** Raman spectra of bulk PP sheet (blue line), a-PP droplets from the surface of standard (std.) PP sheet (red line) and standard amorphous PP wax (pink line), respectively. **i** The GPC detector response as a function of elution time for the a-PP droplets. **j** The MW (red line) and MWD (blue line) of main peak in Fig. 1i. Source data are provided as a Source Data file.

lapsed studies (Suppl. Fig. 10) reveal that phase separation begins at the clamped end and progresses over time along the length of the cantilever, consistent with a flow rate that is controlled by the local stress level. Ultimately, the high density of a-PP droplets at the clamped end underwent surface tension-driven coalescence, which is typical of an amorphous polymer above its glass transition temperature (−20 to −5 °C for a-PP[41][42]).

Using standard beam theory that accounts for the large deflections used in these experiments, the compressive stress $\sigma$ along the cantilever's surface in Fig. 2a–g was calculated to range from 0 at the free end to −7.5 MPa at the clamped end (negative values indicate compressive stress). The surface stress 5 mm from the free end is − −0.71 MPa, a value that corresponds to the threshold or yield stress for a-PP flow at 95 °C. The analysis of the data in Fig. 2a–g summarized in Fig. 2h shows that the volume of a-PP released onto the cantilever per unit area at every location along the cantilever length directly correlates with the stress value at that location. The experiment in Fig. 2a–g was repeated at lower temperatures (85, 78, 70 and 60 °C) where nano and microscale phase separation of a-PP onto each cantilever surface was measured and similar trends to that in Fig. 2h were found in each case. Shortened cantilevers were used to generate higher levels of surface stress (up to 26 MPa) that resulted in enhanced levels of a-PP

droplet formation for all temperatures between 35 °C (typical environmental temperature[22]) and 95 °C (Suppl. Table 1). These experiments were designed to mimic permanent residual stresses in plastics, but any creep or stress relaxation of the cantilever material during measurements will result in stress-driven a-PP flow rates that are lower bounds on the true values.

For each temperature measurement condition, the normalized volumetric flow of a-PP to the surface at different locations along the cantilever length was determined as the total measured volume ($\mu m^3$) of the droplets per unit surface area ($\mu m^2$) per unit time (h). The results for all five temperature conditions are shown in Fig. 2i. It is immediately evident that at each temperature the normalized volumetric flow $\hat{v}$ scales linearly with the level of compressive stress ($R^2$ values 0.92 to 0.99). The intercepts at $\hat{v} = 0$ represent the threshold or yield stress $\sigma_y$ to initiate a-PP flow at the surface. Figure 2j shows that the non-zero yield stress ($\sigma_y$) increases as the temperature is reduced so that higher stresses are required to initiate flow at lower temperatures due to increased stiffness of the a-PP. Moreover, the inset in Fig. 2i shows that the intercepts at $\sigma = 0$ are similar ($\hat{v} \approx -0.5\,\mathrm{nmh}^{-1}$) for all temperatures studied, suggesting a common flow mechanism. Collectively, the results in Fig. 2i are consistent with the a-PP behaving like a Bingham plastic material, which is solid (with zero shear rate or flow) when the

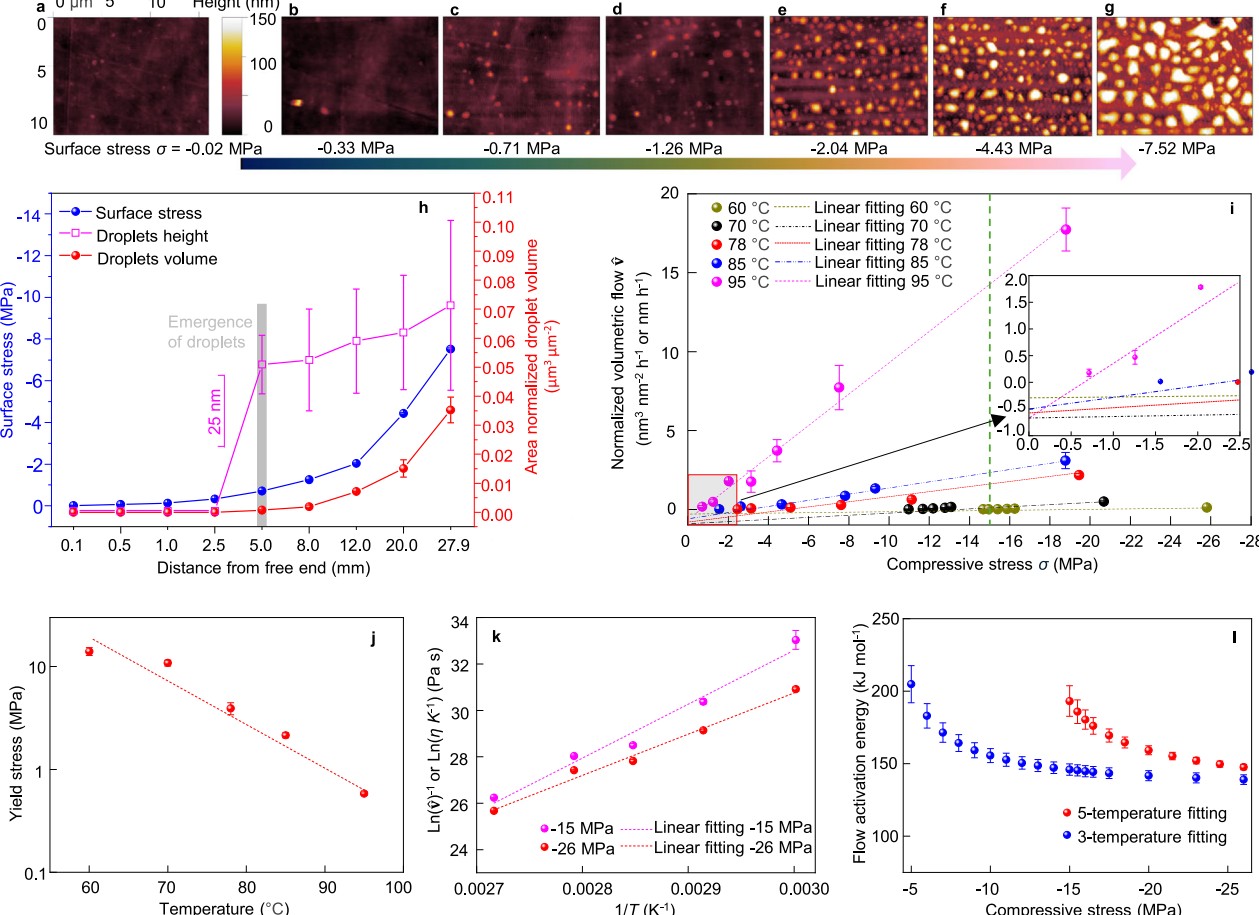

**Fig. 2 | Quantitative analysis of surface stress driven a-PP phase separation.** **a**–**g** AFM images across the compressive surface of PP cantilever after 4 h of 95 °C oven heat. All images have the same scale bars. **h** Surface stress (blue line), a-PP droplet average height (pink line), and a-PP droplet volume per unit (red line) area across the compressive surface of PP cantilever after 4 h of 95 °C oven heat, respectively. **i** The normalized volumetric flow $\hat{v}$ of a-PP under different surface stress conditions and for oven temperatures of 60–95 °C, respectively. The linear fitting is consistent with the a-PP behaving like a Bingham plastic fluid. **j** The yield stress $\sigma_y$ to initiate a-PP flow to the surface of bulk PP sheet under conditions of 60 to 95 °C oven heat, calculated from the intercepts of normalized volumetric flow, $\hat{v} = 0$ in Fig. 2i. **k** Arrhenius plot of Eq. (2) at compressive surface stress conditions of −15 (pink line) to −26 MPa (red line). Data extracted from 5 temperatures in Fig. 2i (green dash line). The slope of linear fit at each stress is the flow activation energy. **l** The flow activation energy obtained from 5-temperature fit (Fig. 2k) and 3-temperature fit (Suppl. Fig. 13). Error bars indicate standard deviation. Source data are provided as a Source Data file.

applied stress $\sigma$ is lower than the threshold yield stress $\sigma_y$ [43]. Simulations of Bingham plastic squeeze-film flow through pores within a compressed stationary matrix (see Methods) yielded a flow rate that is proportional to the local stress, consistent with the behavior in Fig. 2i (see Suppl. Fig. 11).

To describe the temperature-dependent flow we note that the slope of each line in Fig. 2i is related to the viscosity of the a-PP polymer, and differences in slope are consistent with the viscoplastic flow being activated and described by Eq. (1) [42].

$$\eta = A\, e^{(E_f/(RT))}, \qquad (1)$$

where $\eta$ is the viscosity of the fluid (Pa s); $E_f$ is the flow activation energy or barrier (kJ mol⁻¹), a critical parameter for understanding the nature of amorphous polymers[42]; and R is the gas constant. For an inertia-less flow of extruded a-PP, the velocity is expected to vary as the reciprocal of its viscosity so that at a given stress, $\hat{v} = \frac{k}{\eta}$, where $k$ is a constant[43]. On this basis, we can rewrite Eq. (1) as:

$$\ln\left(\frac{1}{\hat{v}}\right) = \ln\left(\frac{A}{k}\right) + \frac{E_f}{R} \times \frac{1}{T} \qquad (2)$$

and use this form to replot the data in Fig. 2i for each of the stress conditions in our experiments. This was accomplished by choosing a particular stress and using the fitted lines in Fig. 2i to estimate the volumetric flow at that stress for each of the measured temperatures (see, for example, the vertical dashed green line in Fig. 2i used to obtain $\hat{v}$ at −15 MPa for each temperature). This process was then repeated for a range of stress conditions, and the data plotted in a $\ln\left(\frac{1}{\hat{v}}\right)$ vs. $\frac{1}{T}$ analysis to determine the energy barrier to flow at each stress. However, since the threshold yield stress for flow $\sigma_y$ increases as the temperature is reduced (see Fig. 2j), it is not possible to perform a single Arrhenius analysis over the entire stress range. Accordingly, the analysis was performed over two stress windows: the high-stress range in Fig. 2i from −15 to −26 MPa for which there is data at all five temperatures, and the full stress range from −5 to −26 MPa where data is available only at the three highest temperatures (78, 85, and 95 °C). The corresponding Arrhenius analyses are shown in Fig. 2k and Suppl. Fig. 12 for representative stresses. The slopes of the Arrhenius plots in Fig. 2k show the barrier to flow is greater at −15 MPa compared to that at −26 MPa, consistent with experiment.

Figure 2l shows the results of the combined Arrhenius analyses plotted as a function of applied stress. There is a reduction in the

energy barrier to flow at higher levels of compressive stress, consistent with the data in Fig. 2a–g, and both analyses show an asymptotic behavior at large stress that approaches a value 143.3 ± 4.3 kJ mol⁻¹. The 5-temperature analysis shows a barrier of up to 200 kJ mol⁻¹ at intermediate stress that is due to the inclusion of low-temperature data. A previous study[44] of the temperature-dependent viscosity of pure a-PP between −7.0 to 70.2 °C showed an energy barrier of 159.0 kJ mol⁻¹ (Suppl. Fig. 13). The larger barrier in our case is likely due to the crystalline matrix that impedes the flow, so that at a given temperature it requires significantly increased levels of compressive stress to generate a flow in semicrystalline PP comparable to that found in pure a-PP.

### Stress-driven release of micropollutants into water

The fate of surface-segregated a-PP depends on local environmental conditions. When the a-PP flow reaches the zero-stress surface the droplets formed become solid Bingham plastics[43,45]. Rheological and GPC studies showed that these low MW droplets are deformable even under modest conditions, suggesting that their release as APMPs into water could be influenced by physical factors, such as air bubbles and turbulence, in parallel with extrinsic environmental factors such as UV-induced degradation or mechanical abrasion. To test for their potential release as APMPs in the presence of water, PP sheets formed into a cylindrical shape (diameter 35 mm and compressive stress of −4 MPa) were soaked in 95 °C DI water for 4 h, mimicking standard testing conditions (EU Regulation 2020/1245 for any plastic-contact condition with temperatures exceeding 40 °C [46], a scenario commonly encountered in plastic plumbing, bottles and containers used in food preparation, Fig. 3a). Under these still conditions there are no shear forces acting on the plastic surface as might occur during water flow or agitation, which would accelerate the detachment of a-PP droplets or traditional polycrystalline PP MPs from the surface. Inspection post-exposure showed that some areas of the compressed inner PP surface (blue box) contained large numbers of a-PP droplets, similar to that found on oven-heated cantilevers, whereas in other areas the droplets were deformed into circular shapes by air bubbles that formed on the surface of the PP sheet during the soak (Fig. 3b). It is well established that capillary forces and pressure shocks exerted by the nucleation, expansion and collapse of air bubbles can re-arrange surface adsorbed particles into ring-like debris fields[47–49]. Once again, no a-PP droplets were found on the outer tensile surface of the cylindrical sheets. An analysis of the water in which the PP sheet was soaked revealed that 208 ± 81 APMPs were released per cm² of the sheet (Fig. 3c, ethanol-pretreatment was employed to eliminate interference from additives[35] and oligomers[9]), compared to only 6 ± 4 semicrystalline PP MPs. In contrast to the rigid and irregularly shaped MPs commonly found following environmental degradation, APMPs have a smooth appearance consistent with their amorphous structure (Fig. 3c). Membrane-captured PP APMPs exhibited a Raman HQI of 0.98 when compared with standard amorphous PP wax (Fig. 3d). Comparison of the Raman spectra of surface segregated a-PP and PP APMPs showed they were near identical (HQI of 0.99), confirming that the a-PP droplets are the source of PP APMPs released into the water. Figure 3e shows the corresponding APMP release from PP sheets with different levels of compressive stress. The modest increase in APMPs released between the −4 MPa and −8 MPa conditions is due to a-PP droplet coalescence (see right-hand panels in Fig. 2f, g), so that release requires a greater force to overcome the increased adhesive footprint and internal friction within droplets, both of which scale with droplet size[50]. Compared to PP sheets, stress-driven flow onto the surface of semicrystalline PE sheets is ~5.5 times more pronounced under the same temperature and compressive stress conditions (Suppl. Fig. 14, Suppl. Note 3). Air bubble-driven rearrangements of the droplets are also more pronounced in the case of PE and result in the formation of larger quantities of circles and circles residuals (Fig. 3b, Suppl. Fig. 14c), indicating

that the specific nature of polymer substantially influences both the flow rate and the interaction of the amorphous polymer with water.

Based on these results and the known level of residual stress (up to −50 MPa[29,51–53]) incorporated into plastic products, the stress-driven release of PP APMPs should be a widespread phenomenon. To test this hypothesis, we filled a brand-new PP bottle (diameter of ≈55 mm-body region, ≈45 mm-mouth/neck regions) with 95 °C DI water and placed it in a 95 °C DI water bath for 4 h, following the standard protocol (EU Regulations[46,54]) (Suppl. Fig. 15). An analysis of the water within the bottle showed that 628 ± 69 PP APMPs were released per cm² of the bottle's interior surface compared to only 0.1 semicrystalline PP MPs per cm² (Fig. 4a–d). This value corresponds to over 300,000 APMPs released per liter and is larger than that released from the smaller diameter cylinder-shaped PP sheet in Fig. 3, which based on its greater curvature, should have higher levels of stress. Although the exact number of amorphous polymer nano-pollutants (APNPs) was not quantified, our previous study[11] demonstrated that much higher levels of PP APNPs passed through the 800 nm membrane pores, shown in Fig. 4a. Scanning electron microscopy (SEM) post analysis identified large numbers of a-PP droplets and circular debris fields on the surface around the inner neck and mouth regions of the bottle (Fig. 4e, f) while there were no obvious a-PP droplets or circles observed on the inner surface of the bottle body (Suppl. Fig. 16). These observations are consistent with the presence of higher stresses in the neck and mouth regions, typically 10 MPa higher than that the body due to clamping during manufacture and its reduced radius of curvature[30,52]. To confirm this, the bottle neck/mouth and body regions were soaked separately and released 2312 ± 939 and 95 ± 83 PP APMPs per cm², respectively (Fig. 4b), demonstrating that the highly stressed neck and mouth regions were predominantly responsible for APMP release.

## Discussion

Here, we demonstrate that residual stress, an intrinsic property of typical semicrystalline plastics, drives nano- and microscale phase separation of short-chain polymers from the bulk onto the surface of the plastic, from where they can be released as APMPs. The polymer type (PP vs. PE), the MWD, the presence of chemical additives and possibly oligomers, and the processing employed to isolate the final product are all important factors in quantifying the denatured state of the APMPs released by the plastic in response to the stress-driven phase separation phenomenon. In all instances, phase separation and APMP release are polymer-dominated, while the levels of chemical additives and/or oligomers present in droplets depend on the solution processing prior to GPC characterization. Ethanol may not effectively remove all oligomers or additives (there are 400 different types[55]), while other common laboratory solvents like chloroform and hexane can completely dissolve the amorphous droplets within 2 h due to the absence of protective crystalline regions found in the bulk semicrystalline plastics[56].

Different responses are also found for the same polymer blended in different ways. PP sheets and PP bottles behave very differently, with the latter showing vastly higher levels of APMP release. It is well known that a-PP is often directly incorporated into semicrystalline PP as a modifier to enhance the impact strength and thermal properties of final products (such as bottles)[57,58]. This incorporation of a-PP not only increases the potential sources of a-PP migration but is known to result in reduced levels of polymer entanglement[57,58], which likely facilitates a-PP migration under stress conditions. The fabrication of plastic bottles also involves additional complexities. Bottles with mouths and screw threads are formed using molding and blowing techniques (where pre-formed shapes may increase in volume several times the original), and mechanical clamps shape the polymer under heat and pressure. The blow rate and original shape are critical factors in controlling stress-aligned crystallite growth to increase strength and barrier properties. These types of processes not only freeze polymer

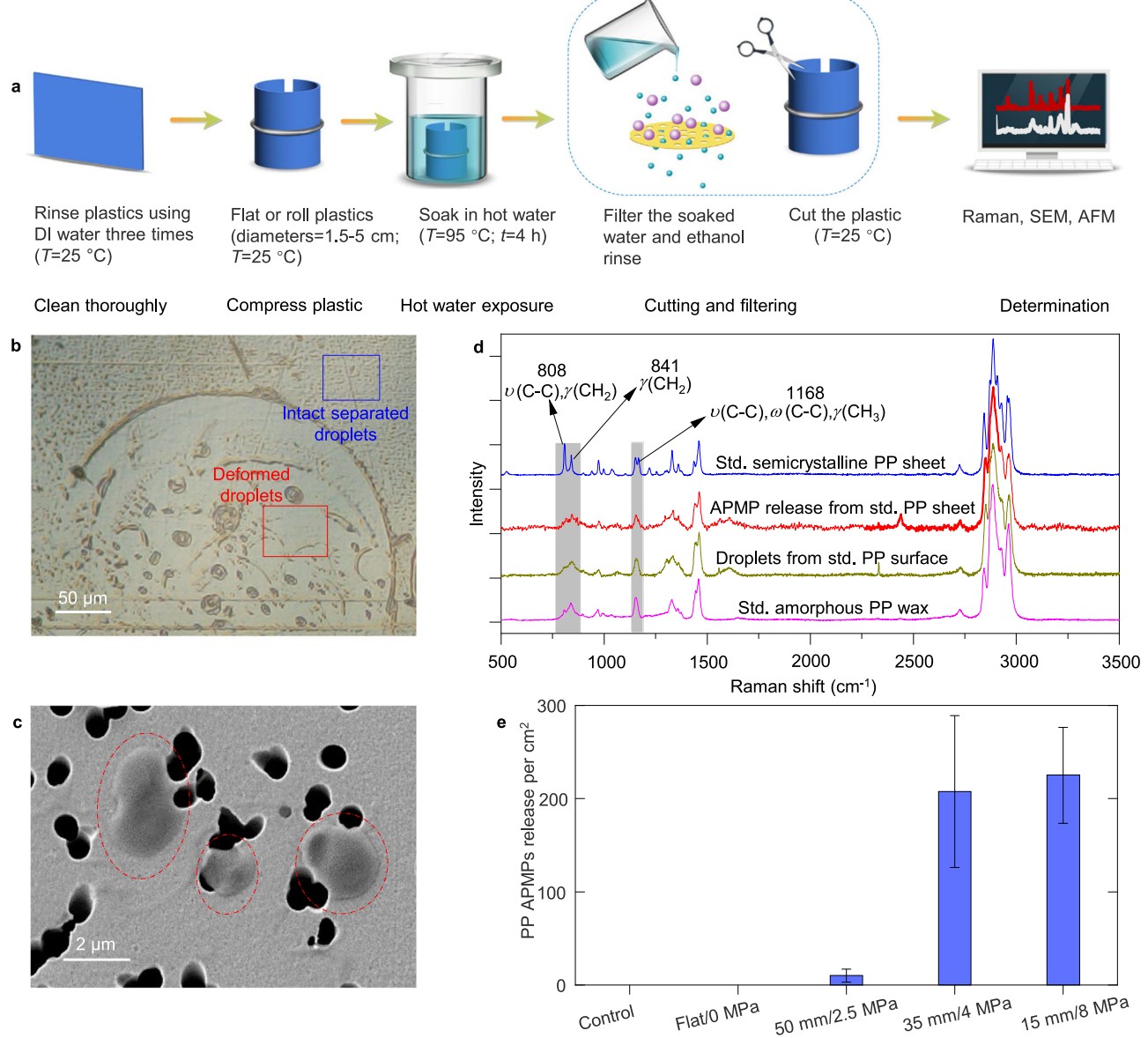

**Fig. 3 | Amorphous polymer micropollutants release from a stressed polypropylene sheet. a** Protocol used to expose stressed PP sheets to 95 °C water for 4 h. **b** Optical image of the inner compressed surface of PP sheet after hot-water exposure. The blue box area has a high quantity of intact a-PP droplets, while the red box showed droplets deformed by air bubble at the hot-water-PP sheet interface. **c** SEM image of typical PP APMPs (red circles) captured on membrane filter surface (800 nm pore size). **d** Raman spectra of typical PP APMPs releases from stressed PP sheet (red line), compared to the spectra from bulk PP sheet (blue line), standard (std.) bulk amorphous PP wax (pink line), and droplets from the surface of the stressed PP sheet (dark yellow line), respectively. **e** The quantity of PP APMPs released from raw flat std PP sheet and stressed PP sheets with different diameters and compressive stresses, respectively. Error bars indicate standard deviation. Source data are provided as a Source Data file.

chains in specific directions but also generate large quantities of surface imperfections[59], potentially altering the stress distribution and migration speed of a-PP[26,60], all of which are consistent with the enhanced APMP release levels noted from the neck region of PP bottles.

Although further research is required, the significant combined global market share of PP and PE (accounts for 54%[31]) underscores the importance of stress-driven phase separation in the plastics industry. The ubiquitous presence of residual compressive stress in plastic products, whether arising from volumetric shrinkage, uneven heat transfer[28,29], or product design, is a fundamental driver of polymer phase separation and APMP generation in water and aquatic environments. Indeed, the same intrinsic factors that control stress-driven phase separation of amorphous polymers may also play a role in the extrinsic environmental degradation of plastics themselves. It is well-known that amorphous polymers are susceptible to oxidation, mechanical abrasion, and heat[20,61], which will likely accelerate the rate of detachment of amorphous droplets from plastics and their release as APMPs into the environment. This is consistent with Raman analysis of PE MPs collected along shorelines that show a distinct absence of the fingerprint crystalline peak at 1416 cm$^{-1}$, confirming their amorphous nature (Suppl. Fig. 14d)[62].

It is important to note that the definition of MPs is rapidly evolving both within the research community and regulatory bodies[63–67]. For instance, ECHA and California Water Boards initially included small-sized semi-solid plastics such as MPs[64,67]. However, ECHA and EU later revised this definition, removing the reference to semi-solids[65,66]. This highlights existing uncertainties in classifying amorphous and semi-

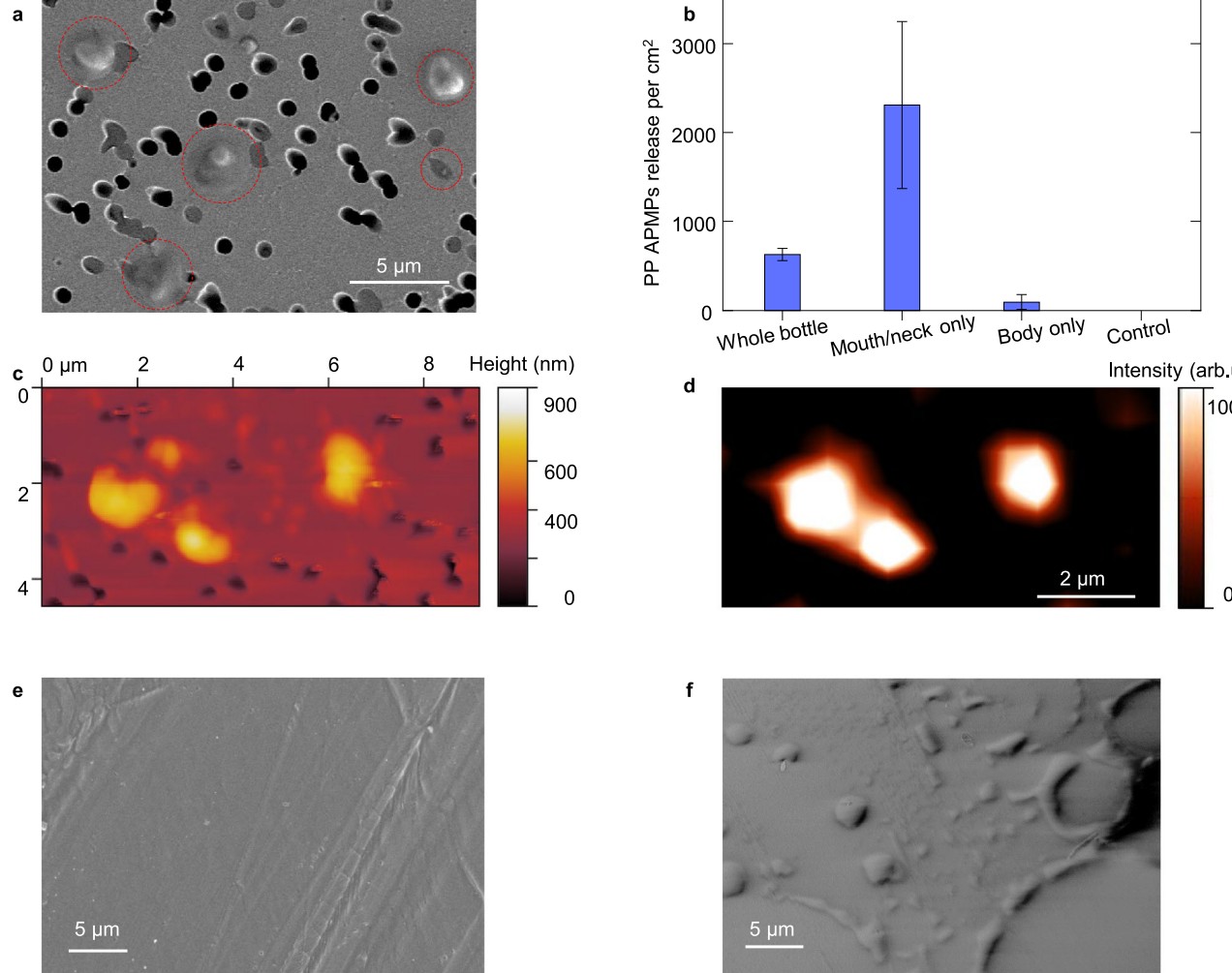

**Fig. 4 | Amorphous polymer micropollutants release from polypropylene bottle. a** SEM image of typical PP APMPs captured on membrane filter surface, marked by red circles. **b** The quantity of PP APMPs releases from the whole bottle, mouth/neck region, and body region, respectively. **c** AFM image of the released solid APMPs as MPs. Their denatured state reduced molecular weight, particles captured on membrane filter surface. **d** Raman mapping and identification of captured APMP particles in Fig. 4c. **e, f** SEM images of the inner surface of bottle's mouth region before and after exposure to 95 °C water, respectively. Error bars indicate standard deviation. Source data are provided as a Source Data file.

solid APMPs as MPs. Their denatured state reduced molecular weight, and amorphous structure compared to the original parent plastic distinguish them from MPs described in the literature[5], which typically have higher molecular weights and are fragments of original polycrystalline plastics. Nonetheless, our observations that plastic degradation under stress conditions results in both structure and compositional changes have strong parallels with degradation in the environment where reduced molecular weight and compositional changes have also been reported, albeit over much longer time scales[68]. Crucially, structure and compositional differences between APMPs and classical MPs may lead to significant variations in the attachment, translocation, accumulation, and associated toxicity whenever APMPs enter the environment or come into contact with biological systems.

The release of denatured plastic micropollutants is a consequence of present-day plastic manufacturing technologies. While various measures may be taken to modify the intrinsic properties of plastics to frustrate phase separation, such as increasing the energy barrier to flow via enhanced entanglement numbers, crystallinity levels, and molecular weights[69–71], there is now an imperative for the industry to modify processing technologies that enable the manufacture of plastic products with significantly reduced levels of residual stress to minimize APMP generation.

## Methods

### Cantilever testing of standard plastic sheets

Rectangular semicrystalline PP sheet (Goodfellow, 40 mm × 3 mm, 0.5 mm thickness, crystallinity degree of ~49.7%) was thoroughly cleaned (DI water, then air-dried) prior to bending into a cantilever by clamping one end and vertically deflecting the other (free end, Fig. 1a, Suppl. Fig. 2). The deflection height, horizontal length and PP sheet length between free end and clamped end were 20, 16, and 28 mm, respectively. The stressed PP cantilevers were then separately placed in ovens at temperatures of 60 to 95 °C for different durations (detailed in Suppl. Table 1). The morphologies of both compressive and tensile surfaces along cantilever PP sheets were examined using atomic force microscope (AFM) before and after heating. Other techniques, including SEM, Raman spectroscopy, and FTIR, were also used to characterize the surface change. Following the same protocol, a piece of standard semicrystalline PE sheet (Alfa Aesar, 20 mm × 3 mm, 1.6 mm thickness, low density) and a shortened PP cantilever (40 mm × 3 mm, 0.5 mm thickness) were also investigated (detailed at Suppl. Table 1). A full factorial design (air, nitrogen, or vacuum; 0 or 18 MPa stress; 35, 60, or 95 °C; 1 or 70 h exposure) was used to systematically evaluate factors affecting phase separation, followed by statistical analyses using a 95% confidence interval (Minitab 22)[72,73].

## Spectroscopic characterization of plastic bulk samples, a-PP droplets and APMPs

A Raman spectrometer (Renishaw InVia) equipped with a 532 nm laser (Coherent Inc.), a cooled charge-coupled device, and a microscope (NT-MDT) with a ×100 objective (Mitutoyo, M Plan Apo) was used to identify the chemical composition. A system calibration using standard silicon wafers was conducted before the sample test. The exposure time for a typical APMP run was set to 10 s with three accumulations. The spectra were obtained in the range of 250–3500 cm⁻¹. WiRE 3.4 (Renishaw) software was used to operate the system and analyze the recorded spectra. To confirm an APMP, the HQI value of 0.7 was set as the threshold during the comparison[35,74]. If necessary, the spectral background of a tested particle was subtracted before analysis. As a complementary spectroscopy to Raman, a diamond ATR (attenuated total reflection) FTIR system (PerkinElmer) was used to further confirm the chemical identity of plastic samples. The detection range was set as 650–4000 cm⁻¹, with a step size of 2 cm⁻¹. After spectroscopy testing, the APMP number released was normalized by filter membrane area, filtrate volume, and total soaked-plastic area.

## Characterization of plastics sheets, a-PP droplets, and APMPs

An NT-MDT AFM, operating with a Nova NT-MDT SPM software) was employed throughout. Raw plastic sheets were thoroughly cleaned using DI water and air-dried before testing. After heat or hot water soaking, plastic sheets were air-dried and tested immediately. Samples were studied using a tapping mode probe (Nanosensors, PPP-NCST), with a scan rate of around 1 Hz, scan size ~30 μm, scan line of 512, tune frequency of ≈160 KHz.

After testing, the data were analyzed with Gwyddion 2.54 software. For each particle, the 3D structures were obtained by 3D view software. Particle dimensions and average heights were estimated using the Profiles analysis. Typical topography maps of separated droplets are shown in Suppl. Fig. 9a.

For SEM analysis (Zeiss Ultra Plus), small pieces (≈0.5 cm²) of plastic sheet samples were cut and cleaned using DI water. After drying, the fixed samples were sputter coated with a 10-nm Pt layer (Cressington 208HR) and analyzed using SEM with an acceleration voltage of 5 kV and magnifications ranging from ×50 to ×10,000. X-ray diffraction (XRD) measurements were performed using a diffractometer (Bruker D8 Discover) equipped with Cu $K_{\alpha 1}$ radiation. Data were collected at room temperature over a scan range of 5° to 35°. After background subtraction, the XRD pattern was delineated into crystalline peaks and an amorphous halo to obtain the degree of crystallinity[75].

GPC (Agilent 1260 Infinity II GPC/SEC) equipped with a refractive index detector was utilized to determine the MW and MWD of the separated droplets[76–78]. The column and detector temperatures were maintained at 30 °C, with a flow rate set at 1.0 mL min⁻¹ and an injection volume of 100 μL. The calibration was performed using polystyrene (PS) standards in the range from 600 to 7,000,000 g mol⁻¹ ($M_W$), enabling the detection of potential oligomers with low MW.

## Determination of the volume of separated droplets on plastic surface

Each droplet can be modeled as an ellipsoid (Suppl. Fig. 9)[79,80]. The volume of each droplet can be obtained using

$$V_i = \frac{\pi}{6} lwh \tag{3}$$

where, $l$, $w$, $h$ is the length, width and height of ellipsoid $i$, respectively. Using the software of ImageJ, $l$ and $w$ can be easily obtained from AFM images. Regarding the height ($h$) (unit of μm), experimentally, it is observed to increase logarithmically with the increase of length $l$,

according to Eq. (4), which was obtained from a cross-section analysis of droplets with different sizes (see Suppl. Fig. 9d).

$$h = 0.0587 \times \ln(l) + 0.0673 \tag{4}$$

where, the unit of 0.0673 is μm. Combining Eqs. (3) and (4), the total volume of all droplets ($V$, unit of μm³) in a particular area of $S$ μm² of the sample can be obtained by adding the volume of each droplet within it (Suppl. Fig. 9a).

Assuming these droplets were obtained after $t$ h of heat, the average volume of droplets separated every hour per unit area of plastic sheet ($\hat{v}$) can be obtained using

$$\hat{v} = \frac{V}{tS} \tag{5}$$

The units of $\hat{v}$ are (μm³ μm⁻² h⁻¹), which is equivalent to (μm h⁻¹). This also can be seen as the normalized volumetric flow of droplets from bulk plastic sheet to the surface.

## Determination of surface stress on standard plastic sheets

The surface stress of a plastic cantilever was calculated from beam theory under large deflection[38,39] (Suppl. Fig. 2b). The compressive stress $\sigma_x$ at horizontal distance $x$ from the free end is:

$$\sigma_x = \frac{Ed(X_0 - x)\sin\alpha}{X_0^2} \tag{6}$$

Where $E$ is Young's modulus of polymer sheet; $d$ is the thickness of plastic sheet; $X_0$ is the whole projected horizontal length between the free end and clamped end; $\alpha$ is the free end angle. Based on Eq. (6), it is convenient to obtain the compressive surface stress across the surface of a plastic cantilever. Evidently, the compressive surface stress is 0 MPa at free end, increasing toward the clamped end. The maximum surface stress at the clamped end was −7.5 MPa at 95 °C for the typical PP setting 1 (Suppl. Table 1).

## Simulation of Bingham plastic amorphous materials separation

The mechanism of amorphous phase flow to the surface of the cantilever beam in the experiment can be analyzed by a simple model involving the squeeze flow of a Bingham plastic material using Wolfram Mathematica software[81,82]. In the compression test, the cantilever experiences high compressive stress at the region close to the clamped end as well as near the surface. Therefore, we consider the scenario where amorphous material is a Bingham plastic, being squeezed out from the crystalline matrix under conditions where the compressive stress is greater than its yield stress. We consider a narrow region of the sample, highlighted by the solid lines in Suppl. Fig. 11a, and interpret them as parallel plates undergoing mechanical squeezing (see schematic diagram in Suppl. Fig. 11). The separation between the parallel plates is roughly the dimension of the protruding droplet areas from the AFM image. The depth of the plates is much less than the cantilever thickness and chosen to represent the approximate uniform levels of compressive stress acting in the near-surface region.

The volumetric flux of the amorphous phase protruding from the surface is proportional to the local compressive stress (which is equivalent to uniform squeezing) and can be described by the simple well-known linear relation[83,84]:

$$Q(y) \propto \frac{H^3 \left(\sigma - \frac{3}{2\epsilon}\sigma_y\right)}{\eta} \tag{7}$$

Where $Q(y)$ is the volumetric flux of the amorphous phase through the cantilever specimen surface, and $\sigma$ is the compressive stress on the

compressed side of the deflected beam? Both depend on the location along the cantilever axis; $H$ is the typical dimension of the channel separation (measured as the dimension of the extruded area on the cantilever); $\epsilon \equiv \frac{H}{L}$ is a geometric parameter under which the lubrication theory is applied: $L$ is the depth of the surface region over which the stress is assumed to be uniform; $\sigma_y$ is the yield stress of the amorphous phase; $\eta$ is the effective viscosity. Equation (7) is the near-Newtonian limit of the relation between the volumetric flux of the material and compressive stress for a Bingham plastic[83,84], as a leading order approximation. The effective viscosity $\eta$ of the amorphous phase is described by Eq. (1).

The compressive stress $\sigma$ can be obtained by the linear elasticity theory under the large deflection condition (Eq. 7). At a given temperature and with a compressive stress $\sigma$ (larger than $\frac{3}{2\epsilon}\tau_0$), the volumetric flux $Q(y)$ linearly increases with the increase of the compressive stress $\sigma_c(y)$, while all other parameters in Eq. (7) are constant. This is consistent with the nature of Bingham fluid[43] and describes the behavior observed experimentally.

### APMPs release from plastic sheets and plastic products

**Precautions to prevent sample contamination.** To avoid potential contamination, all hardware and plastic samples were washed and thoroughly rinsed using DI water (room temperature). Prior to sample preparation, all clean hardware and samples were stored in a clean glass container. During the sample preparation and test process, particle-free nitrile gloves and laboratory coats (100% cotton) were worn. All hardware that came into contact with the samples was made from clean glass (typically Borosilicate glass 3.3). All glassware used in the sample filtration and storage (e.g., glass filter holder and glass petri dishes) was thoroughly cleaned using DI water. The DI water was sourced from a Veolia UltraPure water system. This system includes a Thermo Scientific™ Barnstead™ Nanopure unit with a 0.2 μm absolute final filter for dispensing and monitoring DI water, which ensures the DI water quality (with a conductivity of 1.5 μS cm$^{-1}$ and a resistivity of 18.2 MΩ cm). During the test of APMPs, DI water, and glass beakers were used in control experiments following the same protocol (Fig. 3a, no plastic sample). A control sample was tested for every five experimental samples to assess background APMP concentration in DI water and potential contamination during sample preparation. No PP APMPs were found in the control samples, confirming the reliability of our method.

**APMPs release from stressed plastic sheet.** The pre-cleaned PP sheet was rolled to a cylindrical shape (diameter of 35 mm) using a stainless-steel O-ring and soaked in 95 °C DI water for 4 h (per EU Regulation 2020/1245[46], Fig. 3a). The stressed PP cylinder was then placed on a clean glass plate to cool. After cooling down, the PP cylinder was carefully cut and analyzed using AFM, SEM, Raman spectroscopy, and FTIR. The DI water from the soak was gently shaken and filtered through a gold-coated polycarbonate membrane filter (diameter 25 mm with pore size of 0.8 μm, gold-coated track-etched filter, APC[85,86]). After filtration, an additional 20 mL ethanol (Sigma-Aldrich, ≥99.8%) was filtered through to remove any potential plastic additives, to facilitate the identification of APMPs[35]. The membrane filter was then carefully moved to a clean cover glass and immediately stored in a clean glass petri dish (Brand™, FisherScientific). The particles captured on the membrane filter were characterized using Raman spectroscopy, FTIR, AFM, and SEM. Additionally, we strictly followed the regulation[46,54] and tested APMPs release from PP sheets soaked in 100 °C boiling water for 4 h and observed similar morphology changes but recorded much higher levels of a-PP droplets, circular debris fields, and APMP release.

**APMPs release from PP bottles.** Brand-new PP bottles were thoroughly rinsed, filled with 95 °C DI water, and maintained in a 95 °C

water bath for 4 h (Suppl. Fig. 15). The bottle was then moved to a clean glass plate to cool. After cooling down, the water in the bottle was filtered following the same protocol used for the stressed plastic sheet. The particles captured on the membrane filter were characterized using Raman spectroscopy, FTIR, AFM, and SEM. Post-analysis of the soaked bottle involved carefully cutting it into small pieces for analysis by AFM, SEM, Raman spectroscopy, and FTIR to observe the surface morphology and changes in chemical composition. To investigate APMP release from the bottle's body region, the water volume was reduced to prevent contact between the bottle's neck/mouth and the water surface. After this, the PP bottle was examined using the same protocol. Similarly, during investigations of APMPs release from the neck/mouth region, the bottle was inverted and immersed just beyond the neck region to avoid contact between the bottle body and the water.

## Data availability
The data that support the findings of this study are available from the corresponding author upon request. Source data are provided with this paper.

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

## Acknowledgements

This work was supported by Science Foundation Ireland grants 20/FIP/PL/8733, 12/RC/2278_P2 and 16/IA/4462 (J.J.B., J.J.W.). Research Ireland grant 24/PATH-S/12599 (D.L., J.J.B.). Enterprise Ireland grant CF-2021-1729-1 (L.X., D.L.). Enterprise Ireland grants CF-20180870 (J.J.W., L.X.). Advanced Talents Project by Zhejiang A&F University RC2024F01 (D.L.). We appreciate the professional help from technician teams of Trinity Civil, Structural and Environmental Department (David A. McAulay, Mark Gilligan, Mary O'Shea, Patrick L.K. Veale, Robert Fitzpatrick, etc.), Photonics Laboratory and AML in CRANN/AMBER Research Center.

## Author contributions

J.J.B. led the overall effort. Conceptualization and supervision: J.J.B., L.X., J.J.W., D.L.; methodology and data collection: Y.S., E.D.S., Z.Z., L.Y., C.H., C.G., M.R., J.P.; stress determination and simulation: P.L., J.E.S.; writing-original draft: D.L., J.J.B., P.L.; writing-review and editing: J.J.B., L.X., J.J.W., J.E.S., M.A.M. All authors discussed the results and commented on the manuscript.

## Competing interests

The authors declare no competing interests.
