## [Transparent Peer Review file · Nature Communications]

Stress induced phase separation in plastics drives the release of amorphous polymer micro-pollutants into water

Corresponding Author: Dr John Boland

Version 0:

Reviewer comments:

Reviewer #1

(Remarks to the Author)

Stress drives polymer phase separation 1 and microplastic release into water

General:

Interesting work!

Methods and interpretations of the data appear robust.

Study offers novel insight into a mechanism by which nano- and micro-plastics form during the manufacture and use of plastic products – i.e. physical, compressive stress driven exclusion of amorphous PP to polymer surfaces. Aspects of the work could be better introduced. The abstract and end of the introduction are places where this aspect could be improved. The introduction should also introduce sources of compressive stress in real world settings, plus define and also describe the levels of residual stress in real world settings. This will better prime a reader to understand the importance of the work while reading.

It would also be important for the authors to include a statement or discussion of whether other processes may also lead to the prevalence of a-PP microplastics and nanoplastics they observe being released from the water bottle.

Discussion and introduction should also note there is wealth of studies showing other pathways to MP formation in the environment from processes such as photochemical degradation, abrasion, etc. No specific paper recommended - quick scholar search yields 10s of thousands. Read a few reviews and cite a favorite along with core processes that are understood to yield nano- and microplastics in the environment. This will go along with revising the "no fundamental" knowledge statement noted below.

Abstract:

Needs significant editing to understand what was done and the implications.

Line 23: saying there is "no fundamental understanding" is incorrect. Incomplete, gaps in the .. etc would be more correct.

Line 25: Global market – maybe "global plastics production" if that is the meaning.

Line 25: Residual stress from what and of what type? Abstract needs work to be intelligible to broader audience.

Line 27: where do the droplets come from – internally and forced out? Needs to be clear when reading the abstract without reading the paper.

Line 27: Simulations? Lab? Computer? What type of simulation?

Lines 29-31: Are these two statements research findings? As written, they just seem like statements of fact that was know prior to your work.

Line 32: What were these studies with bottles? Were they physically testing the bottles in the lab? Making the bottles?

Analyzing pre-made bottles in the lab? Computer simulations of perceived stress during manufacturing?

Line 39: Many drinking bottles are PET. Caps are often PP. PP is pretty brittle for water storage.

Line 48: "no fundamental" is too broad. There is some understanding. Just not complete.

Line 56: define residual stress.

Section starting line 59: Add more about the methods employed – e.g., if physical or computational in each case and map out the flow of the study to better introduce the reader to what you are doing and why.

Line 71: You “began” as the study was done in the past.

Line 73: How was vertical deflection attained?

Line 120: note the glass transition temp for a-PP

Line 180: “the presence of the crystalline matrix through which the a-PP must pass through increases the barrier to a-PP flow” - unless you have direct evidence of this as the cause, then it is best to state it as a potential explanation.

Lines 173 to 182: what were the errors on your and the past estimate of the energy barrier to a-PP flow? Important to report the uncertainty and, in this case, to verify that your value of ~140 is significantly different to the previously reported value of 159.

Line 234: Present PE in as much detail as PP or explain why you do not do this in the main text.

Review by Aron Stubbins

Reviewer #2

(Remarks to the Author)

I find this research about the generation mechanisms of nano/microplastics at the surface of PP/PE fascinating and extremely timely and needed – the authors correctly state that we have a major global challenge, but we understand too little on the fundamental mechanisms.

First, I want to explicitly state that I am not polymer materials scientist and therefore I am possibly not the best expert to assess the detail of the research here. The results are very noteworthy and intriguing. However, based on my wider expertise, I have some reservations on specific aspects and/or suggestions for improvement.

On the methodology: research includes a long series of experiments trying to quantify the response of the generation of formations on the surface of the plastic under a variety of conditions: Stress configuration settings (cantilever set up dimensions), temperature, time of exposure, other ambient conditions (nitrogen, vacuum and water) for both PE and PP. While the results are extensively reported in informative figures, what I miss is an over statistical design of experiments (DoE) that would allow a more wholistic testing a reporting of the effects here and in particular maybe provide insights on combined (confounded) effects. This is a standard practice. Maybe the tests have been conducted in a fashion that such a data analysis is also feasible?

There are some extreme settings here, especially for the temperature range 35-95 °C. Not sure under which conditions the high range of temperature is of relevance here? Maybe in a desert setting? One thing is to try extreme temperatures to reveal the phenomena involved – and a different thing to establish relevance of plastic pollution problems. Especially, for example, when some of the effects are evident only at elevated temperatures: In Fig 2c: little /no effect is evident at 60 °C and below. Therefore, the relevance of the settings to nano/microplastics formation should be discussed. I might not have read carefully, but I do not fully understand the experimental setting on the temperature side. The standard sequence of stressing and then heating – does the sequence of applying heating then stressing been tested for? How this simulates the formation of the plastics, when I guess they are first heated and then stressed). How does this differentiation affect the testing relevance here?

At these elevated temperatures (95 °C), there might be other phenomena that could be taking place.

Regarding the relevance to the wider plastic pollution what I miss from the discussion is how these nano- and micro plastics similar in any way to those documented in the literature from lab tests and in environmental/ field studies. Or there is nothing to compare with?

The authors the associate the formation of amorphous nano plastics on the surface of the polymer. Is there any physical explanation on the nature of the flow via the crystalline layers that sandwich the amorphous ones? Why this is predominantly an effect of stress and not elevated temperature? This is what I personally conclude from the Figure 2 C – or at least that there is a step change in the behaviour (effect on normalised volumetric flow) from 85 to 95 °C: this is where a 3D plot of response surfaces may be helpful and DoE to assess and the comparative magnitude of the effect could help. So not fully convinced that the term “drives” is scientifically accurate in the title. This expression implies that this is the main mechanism and the variable that controls this mechanism, whilst it seems to me that temperature and phenomena associated with (driven by?) its change are clearly more potent here.

There seems to be less controlled experimental settings with testes pertaining to release of the formulated nano-microplastics on the polymer surface into the water. I am not sure the phenomena around hot water of (95 °C) are accounted for and can only be associated with what it was tested in the atmospheric environment.

The authors conclude: “Our findings underscore the need to modify current plastic processing technologies to reduce

residual stress levels and suppress amorphous phase separation and microplastic release.” However, I am not sure how much of the potential for formulation of nano/microplastics could be mitigated by just washing the products in the first place? Also, a grey area remains establishing a better equivalence between the remaining stress in commercially produced plastics and the laboratory settings of the tests here. Maybe the authors could elaborate further in that.

Despite these generic questions, my opinion is that the research reported here is very useful and critically advances our understanding. Maybe it could benefit from an overall more open discussion on its limitations, also referring to its relevance to actual average ambient conditions under which microplastics are generated.

Reviewer #3

(Remarks to the Author)

The manuscript “Stress drives polymer phase separation and microplastic release into water” presents a study on the formation of microplastic on the surface of polypropylene (PP) and its release into water. The topic seems current and important, even though the problem of microplastics has existed for several decades without our awareness.

The performed tests are interesting; however, the presented interpretation of the data is fundamentally incorrect. According to the author’s interpretation, compressive stress and elevated temperature drive the nano- and microscale segregation of amorphous polymer droplets onto the surface of polypropylene. The analyzed polymer is a semicrystalline material. During solidification, its macromolecules are incorporated into the lamellar crystals. It is therefore not possible for macromolecules to be released from the crystals (without melting the crystals), disentangle from the molecular network of amorphous regions, and finally diffuse/migrate to the surface of the material, creating the mentioned domains, even at elevated temperature and under compressive stress. Even assuming that such migration of PP molecules takes place, why do they not crystallize within the droplets as demonstrated by spectroscopic techniques?

In my opinion, elevated temperature and compressive stress drive the migration of propylene oligomers and the formation of their droplets on the surface of the material. Initially, during the solidification process, the oligomers are expelled from growing crystals into intercrystalline regions. Due to the significantly smaller dimensions of the oligomer molecules compared to the PP molecules, their migration through amorphous regions to the material surface is more probable. This process is significantly accelerated at elevated temperatures. Moreover, the presence of compressive stresses further enhances the efficiency of oligomer migration to the material surface by decreasing the thickness of interlamellar regions (as schematically presented in Fig. S8c). Finally, conditioning PP samples at elevated temperatures enhances the degree of crystallinity of the polymer matrix. The reduced content of amorphous phase restricts the available “space” for oligomers, leading to their outflow from the interlamellar regions towards the material surface. It is also worth noting that the oligomers present in commercially available polypropylenes are chemically similar to polypropylene (PP) matrix or amorphous PP wax (in fact the atactic PP). Additionally, these oligomers practically do not crystallize (European Polymer Journal, 2015, 69, pp. 186–200, 6938), similar to the material found in the analyzed droplets.

To sum up, this manuscript observes and analyzes the process of oligomer migration, which is an issue addressed in other works (e.g., Foods 2020, 9(10), 1365), rather than focusing on the formation and release of microplastics.

Other remarks.

No ethanol is used to remove non-polymeric substances (additives, oligomers) from polyolefins (PP or PE). Typically, a mixture of hexane, chloroform, and ethanol (Polym. Degrad. Stab. 1998, 60, 137–143; Polym. Degrad. Stab. 2006, 91, 1598–1605) is used for effective removal of such substances.

There is a lack of basic information about the materials analyzed in the work. How was the degree of crystallinity determined?

I do not recommend the manuscript for publication because of the incorrect interpretation of experimental data and the lack of novelty.

Version 1:

Reviewer comments:

Reviewer #3

(Remarks to the Author)

This is a review of the work titled: Intrinsic Stress Drives Polymer Phase Separation and Microplastic Release into Water.

The test performed using the GPC technique clearly demonstrated that the molecular weight (MW) of the main fraction of substances in the droplets is five times lower than the average MW of the analyzed polypropylene. Additionally, the lack of crystallization ability is a manifestation of insufficient regioregularity (isotacticity) of the chains, similar to atactic polypropylene wax. Therefore, from both a physical and chemical perspective, the material in the droplets has completely different properties compared to the polypropylene analyzed in the manuscript. Due to the lower MW, lack of crystallinity, and regioregularity of the chains, the glass transition temperature (T_g) of this amorphous mixture is likely shifted 20–30°C lower compared to the T_g of polypropylene. Thus, at room temperature, the material within the droplets exhibits a waxy consistency, akin to atactic polypropylene wax.

According to the generally accepted definitions (U.S. National Oceanic and Atmospheric Administration, U.S. Environmental Protection Agency, or European Chemicals Agency), microplastics are small plastic pieces or particles less than five millimeters in size.... The term 'pieces or particles' refers to objects that are solid under given (environmental) conditions. Therefore, the use of the term 'microplastic' in relation to droplets of substances (with inhomogeneous content) that have a gooey or waxy consistency seems unjustified.

The experiment, schematically presented in Figure 4R, does not practically contribute or explain anything significant. Since the authors observe the migration of a molecular fraction with a mass of 22,000 g/mol, what more could have been expected in the case of a material with a molecular weight half that size?

In relation to the test presented in Fig. R5: In the original review, I postulated an increase in the degree of crystallinity of the polypropylene matrix, not a change in the crystallinity of the material within the droplets due to the annealing process at 95°C.

Accordingly, the article may be accepted for publication if the authors remove the word 'microplastic' from the title and content of the work

Version 2:

Reviewer comments:

Reviewer #3

(Remarks to the Author)

Unfortunately, I cannot agree with the interpretation that the material in droplets should be classified as a solid. According to the provided definitions and their interpretation in the context of a-PP wax, a solid should exhibit a melting temperature above 20°C. The authors assert that the material in droplets melts at a temperature above 20°C. The fact that domains are visible in Figure 3b does not prove that the material is solid; droplets of a viscous liquid on a substrate/glass would appear similarly.

Additionally, I am curious about the authors' interpretation of the nature of the melting process observed above 20°C for the material in droplets. For semicrystalline PP, one would expect crystal melting in the range of approximately 160–170°C. Earlier, in the range of -20 to 0°C, the glass transition temperature of amorphous regions (T_g) is typically observed. In the case of a-PP wax, we would observe only a softening temperature, likely shifted by 10–20°C toward lower temperatures compared to the T_g of semicrystalline material.

Based on the above, the material in droplets is more likely a highly viscous liquid rather than a solid. I therefore maintain my previous position that the use of the term "microplastic" in reference to droplets with a wax-like consistency is unwarranted and may mislead readers regarding the actual source of microplastics present in the environment.

I therefore cannot recommend the publication of the manuscript in its current form, especially in a journal with such a significant impact on the scientific community.

Version 3:

Reviewer comments:

Reviewer #4

(Remarks to the Author)

Review:

Intrinsic stress induced phase separation of low molecular weight polymer from plastics drives the release of amorphous globular microplastics into water
Dunzhu Li et al.

The manuscript is timely, interesting and provides novel results on the degradation of PP and PE.

1. At this point, I would like to mention that the manuscript focuses mainly on PP and PE is only mentioned once in the results section which seems to me an imbalance in the presentation of the results.

2. term "microplastics"

The authors state that the "droplets are sticky gel-like substances", which by definition are semi-solid substances. In the rebuttal, they also claim that, according to their interpretation of the current definitions, these droplets should be referred to as microplastics.

According to the EU 2023 (https://ec.europa.eu/commission/presscorner/detail/en/ip_23_4581) microplastics are defined as follows: The adopted restriction uses a broad definition of microplastics – it covers all synthetic polymer particles below five millimetres that are organic, insoluble and resist degradation.

The adopted Annex XVII reads as follows (2023 Annex XVII to Regulation (EC) No 1907/2006 of the European Parliament

and of the Council concerning the Registration, Evaluation, Authorisation and Restriction of Chemicals (REACH) as regards synthetic polymer microparticles):

Synthetic polymer microparticles:

polymers that are solid and which fulfil both of the following conditions:

(a) are contained in particles and constitute at least 1 % by weight of those particles; or build a continuous surface coating on particles;

(b) at least 1 % by weight of the particles referred to in point (a)

fulfil either of the following conditions:

(i) all dimensions of the particles are equal to or less than 5 mm;

(ii) the length of the particles is equal to or less than 15 mm and their length to diameter ratio is greater than 3.

The following polymers are excluded from this designation:

(a) polymers that are the result of a polymerisation process that has taken place in nature, independently of the process through which they have been extracted, which are not chemically modified substances;

(b) polymers that are degradable as proved in accordance with Appendix [X];

(c) polymers that have a solubility greater than 2 g/L as proved in accordance with Appendix [Y];

(d) polymers that do not contain carbon atoms in their chemical structure.

This roughly corresponds to the definition of microplastics by the International Organization for Standardization (ISO) ISO 24187:2023 (<https://www.iso.org/obp/ui/en/#iso:std:iso:24187:ed-1:v1:en>):

3.2

microplastic

any solid plastic particle insoluble in water with dimension between 1 µm and 1 000 µm (= 1 mm)

Note 1 to entry: Primary microplastics object represents a particle intentionally added to end-user products for example cosmetic means, coatings, paints etc. Secondary microplastics object can also result as a fragment of the respective item.

Note 2 to entry: Microplastics have regular and irregular shapes (see ISO 9276-6:2008).

Note 3 to entry: The defined dimension is related to the longest length of the particle.

[SOURCE:ISO/TR 21960:2020, 3.9, modified — Note 1 to entry was removed, all other Notes to entry were changed.]

In their response, the authors cited an older ECHA definition (2018) that still referred to semi-solid particles, which is no longer the case in the new definition. However, they further clarified how ECHA defines a solid: a substance or mixture that does not meet the definitions of a liquid or gas. This definition is still included in Annex XVII.

However, the analytical research and definition of MP is still an ongoing process and most studies on environmental MP pollution exclude waxes.

One aspect that is also included in the definition of microplastics is the long-term persistence of the particles. No information is given here about the PP droplets, so this criterion can hardly be assessed. I assume that the droplets, due to their amorphous character, are likely to be rapidly degraded by microorganisms in the environment.

Given all these uncertainties, I would rather support reviewer3's opinion not to use the term microplastics, since PP droplets are semi-solid and not solid and have completely different properties (as pointed out by reviewer3 - lower MW, lack of crystallinity, and regioregularity of the chains) than "classic" microplastic particles. In almost all reports on environmental contamination, MPs are classified into fragments, foam, spheres or fibres. None of these properties fits gel-like PP-droplets.

I therefore recommend not to categorize the results found here under the general term "microplastics". I think the authors' findings are new and therefore this degradation product also deserves a new name.

I further recommend avoiding the term "microplastics" throughout the manuscript (in the title, abstract and main text body) to describe gel-like PP and PE droplets.

Further, I strongly recommend not using the term "globular microplastics" (GMP) because "globular" merely means "spherical" and the droplets have nothing in common with spherical MP beads.

Since gel-like PP and PE droplets appear to be a special form of plastic degradation products, which are not considered in the MNP research community, I would rather recommend discussing similarities, but especially differences, to "classic" microplastics, but only in the discussion section.

3. I have an additional concern regarding the methods used:

Methods: Microplastics release from stressed plastic sheet and from PP bottles

The authors write: PP sheets and stainless-steel O-rings were thoroughly rinsed by DI water (repeated 3 times at room temperature)

Was the DI water filtered before use to avoid contamination since DI water usually contains MP?

Which other QA/QC measures were taken to avoid contamination? Were any blanks taken at each sample processing step to test for contamination. This would be crucial to exclude that the measured particles and droplets originate from the samples and not from, for example, air contamination. Although the a-PP droplets can be distinguished from "classic" PP-

MP, contamination of the sample with a-PP droplets cannot be excluded if the degradation mechanism shown here occurs frequently.

Therefore, quality assurance/quality control measures must be reported, otherwise the reliability of the results cannot be fully assessed.

4. minor aspects:

“Water environment” is unusual wording – it is either “in contact with water” or “aquatic environment”

“suggesting that direct release of MPs induced by aquatic factors such as air bubbles is likely to occur in parallel with extrinsic environmental factors such as UV-induced degradation or mechanical abrasion.”

Aquatic factor is not precise. Environmental factors can either be biotic or abiotic, and even abiotic factors, which are probably addressed here, are quite diverse and I think that only physical factors such as turbulence may lead to the release.

Although the exact quantity of nanoplastics was not quantified
I would recommend using “exact number” instead of quantity

Version 4:

Reviewer comments:

Reviewer #4

(Remarks to the Author)

The term plastic micro-pollutants (PMPs) is not very common, but it is used as a synonym for “classical” microplastics. The authors also use the term in this sense (“Micro-pollutants released from semicrystalline plastics are known to threaten the environment and human health “), so they do not draw any conceptual distinction between microplastics and their newly discovered amorphous polymer droplets. I would therefore strongly advise to distinguish between “classical” microplastics and amorphous polymer droplets (APD) in the manuscript and therefore to use the term APD in the title and in the manuscript.

Responses to the referees

We would like to thank the editor and the reviewers for reviewing our manuscript. We agree with most of the comments and the manuscript has been revised accordingly. All changes are highlighted in blue and can be found in the word document (main text-blue tracked.doc). Our responses to the specific comments of the editor and reviewers are summarized below.

Reviewer #1:

Stress drives polymer phase separation 1 and microplastic release into water

1.General:

Interesting work!

Methods and interpretations of the data appear robust.

Study offers novel insight into a mechanism by which nano- and micro-plastics form during the manufacture and use of plastic products – i.e. physical, compressive stress driven exclusion of amorphous PP to polymer surfaces. Aspects of the work could be better introduced. The abstract and end of the introduction are places where this aspect could be improved. The introduction should also introduce sources of compressive stress in real world settings, plus define and also describe the levels of residual stress in real world settings. This will better prime a reader to understand the importance of the work while reading.

Response: Many thanks for your review. We appreciate your kind comments. As suggested, we have revised the Introduction and Abstract to clearly state the sources, definition and levels of residual stress to highlight the pervasive presence of residual stress.

Added text: Fabrication of non-fiber based plastics involves cooling molten resins containing blended polymers and additives into desired shapes, which introduces the residual stresses that are pervasively found in plastic products^{1,2}. Residual stress refers to the locked-in stress distribution present in a structure, component, plate or sheet, while there is no external loads or forces applied². The generation of high levels of residual surface stress is affected by multiple factors such as volumetric shrinkage, flow induced stresses, crystallization and heat transfer^{3,4}. For example, differential cooling and heat transfer cause the surface region of the resin to cool more quickly and to shrink relative to its interior, resulting in compressive stresses at the surfaces of plastic parts, typically 10 to 30 MPa⁴. Even higher levels of residual stress are generated whenever plastic parts are engineered into particular shapes, such as the neck and mouth regions of plastic bottles⁵.

References:

- 1 Guevara-Morales, A. & Figueroa-López, U. Residual stresses in injection molded products. *Journal of materials science* 49, 4399-4415 (2014).
- 2 Schijve, J. *Fatigue of Structures and Materials*. 2nd edition, pp 89 (Springer Dordrecht, 2009).
- 3 Spina, R., Spekowius, M., Dahlmann, R. & Hopmann, C. Analysis of polymer crystallization and residual stresses in injection molded parts. *International journal of precision engineering and manufacturing* 15, 89-96 (2014).
- 4 Mills, N. Residual stresses in plastics, rapidly cooled from the melt, and their relief by sectioning. *Journal of Materials Science* 17, 558-574 (1982).
- 5 Cho, S.-H., Hong, J.-S. & Lyu, M.-Y. Investigation of the molding conditions to minimize residual stress and shrinkage in injection molded preform of PET bottle. *Polymer (Korea)* 35, 467-471 (2011).

Please see the revised text as it appears on Page 1 Lines 24-25, Line 26 and Page 2 Lines 59-65, and Line 67.

2. It would also be important for the authors to include a statement or discussion of whether other processes may also lead to the prevalence of a-PP microplastics and nanoplastics they observe being released from the water bottle.

Response: Many thanks for your constructive comment. As suggested, we have conducted a comprehensive literature review to identify the factors impacting a-PP migration and release. We have incorporated a new paragraph in the discussion section of our manuscript to clearly outline additional factors that influence a-PP migration and release.

Added discussion: Different responses are also found for the same polymer blended in different ways. PP sheets and PP bottles behave very differently, with the latter showing vastly higher levels of MP release. It is well known that a-PP is often directly incorporated into semicrystalline PP as a modifier to enhance the impact strength and thermal properties of final products such as bottles^{6,7}. This incorporation of a-PP not only increases the potential sources of a-PP migration but is known to result in reduced levels of polymer entanglement^{6,7}, which likely facilitates a-PP migration under stress conditions. The fabrication of plastic bottles also involves additional complexities. Bottles with mouths and screw threads are formed using moulding and blowing (where pre-formed shapes may increase in volume several times the original) techniques and mechanical clamps shape the polymer under heat and pressure. Often the blowing and former shape are critical in defining stress aligned crystallite directional growth to increase strength and barrier properties. These type of processes not only freezes polymer chains in specific directions but also generates large quantities of surface imperfections⁸, potentially altering the stress distribution and migration speed of a-PP^{1,9}, all of which is consistent with the enhanced MP release levels noted from the neck region of PP bottles.

References:

- 1 Guevara-Morales, A. & Figueroa-López, U. Residual stresses in injection molded products. *Journal of materials science* 49, 4399-4415 (2014).
- 6 Nam, B.-K., Park, O. O. & Kim, S.-C. Properties of isotactic polypropylene/atactic polypropylene blends. *Macromolecular Research* 23, 809-813 (2015).
- 7 Chen, J.-H., Tsai, F.-C., Nien, Y.-H. & Yeh, P.-H. Isothermal crystallization of isotactic polypropylene blended with low molecular weight atactic polypropylene. Part I. Thermal properties and morphology development. *Polymer* 46, 5680-5688 (2005).
- 8 Pedgley, O., Şener, B., Lilley, D. & Bridgens, B. Embracing material surface imperfections in product design. *International Journal of Design* (2018).
- 9 Syrett, D. Bottle design and manufacture and related packaging. *Carbonated Soft Drinks: Formulation and Manufacture*, 181-217 (2006).

Please see the revised text as it appears on Page 7 Lines 289-303 and SI Page 11 Figure S8.

3. Discussion and introduction should also note there is wealth of studies showing other pathways to MP formation in the environment from processes such as photochemical degradation, abrasion, etc. No specific paper recommended - quick scholar search yields 10s of thousands. Read a few reviews and cite a favorite along with core processes that are understood to yield nano- and microplastics in the environment. This will go along with revising the "no fundamental" knowledge statement noted below.

Line 23: saying there is "no fundamental understanding" is incorrect. Incomplete, gaps in the .. etc would be more correct.

Response: As suggested, we have revised the text to clearly outline previous research on microplastic release mechanisms and to emphasise that this research focuses on the intrinsic stress level in plastics rather than the response of plastics to extrinsic factors such as weathering or UV exposure. Additionally, we have accurately identified the research gaps in both the Introduction and Abstract to better frame our study's contributions to the field.

Added text: Despite the myriad of potential sources, most studies have focused on MP release mechanisms induced by extrinsic factors, such as UV irradiation and/or mechanical abrasion¹⁰⁻¹². Currently, there is no fundamental understanding of the intrinsic factors such as residual stress that are involved in the generation of MPs.

References:

- 10 Andrady, A. L. The plastic in microplastics: A review. *Marine Pollution Bulletin* 119, 12-22 (2017).
- 11 Shi, Y. et al. Formation of Nano-and Microplastics and Dissolved Chemicals During Photodegradation of Polyester Base Fabrics with Polyurethane Coating. *Environmental science & technology* 57, 1894-1906 (2023).
- 12 Song, Y. K. et al. Combined effects of UV exposure duration and mechanical abrasion on microplastic fragmentation by polymer type. *Environmental science & technology* 51, 4368-4376 (2017).

Please see the revised text as it appears on Page 1 Lines 24-25 and Page 2 Lines 52-55.

4. Abstract:

Needs significant editing to understand what was done and the implications.

Line 25: Global market – maybe “global plastics production” if that is the meaning.

Response: As suggested, we have thoroughly revised the entire abstract to clearly delineate the research gaps identified in previous studies, along with the results and implications of our current study. To improve clarity, the term 'global market' has been updated to 'global plastics production' in both the abstract and the introduction.

Please see the revised text as it appears on Page 1 Line 26, Lines 36-38 and Page 2 Lines 75.

5. Line 25: Residual stress from what and of what type? Abstract needs work to be intelligible to broader audience.

Response: Residual stress generation stems from multiple factors, including volumetric shrinkage, flow-induced stresses, crystallization and heat transfer. These stresses are categorized as processing-induced residual stresses. As suggested, we have revised the Abstract and Introduction to enhance readability for a broader audience.

Please see the revised text as it appears on Page 1 Line 27 and Page 2 Lines 59-65.

6. Line 27: where do the droplets come from – internally and forced out? Needs to be clear when reading the abstract without reading the paper.

Response: Yes, they come from within the bulk plastic itself. Compressive stress forces amorphous PP droplets onto the surface of the semicrystalline plastic. As suggested, we revised the sentence as below to enhance the clarity of abstract.

Revised sentence: We demonstrated that compressive stress forces internal nano- and microscale segregation of amorphous polymer droplets onto the plastic's surface.

Please see the revised text as it appears on Page 1 Line 28.

7. Line 27: Simulations? Lab? Computer? What type of simulation?

Response: It is the Squeeze Flow Model simulated using Wolfram Mathematica software. As suggested, it was revised as *Squeeze Flow Simulation* in Abstract. Other information was added in Method section.

Please see the revised text as it appears on Page 1 Lines 29 and Page 11 Lines 4442-443.

8. Lines 29-31: Are these two statements research findings? As written, they just seem like statements of fact that was know prior to your work.

Response: They are the findings of this research. As suggested, we revised the sentence as below to enhance the clarity of abstract.

Revised sentence: We confirmed that flow is thermally activated and stress dependent, with a reduced energy barrier at higher compressive stresses.

Please see the revised text as it appears on Page 1 Line 31.

9. Line 32: What were these studies with bottles? Were they physically testing the bottles in the lab? Making the bottles? Analyzing pre-made bottles in the lab? Computer simulations of perceived stress during manufacturing?

Response: We conducted the lab experiments on commercially available plastic bottles to demonstrate the phase separation and microplastic release in real daily life, confirming that our finding on standard PP plastic sheets is a widespread phenomenon. The bottles were randomly purchased in local stores. Before test, brand-new PP bottles were thoroughly rinsed using DI water (repeated 3 times at room temperature). The residual stress level and distribution on the plastic bottle were well characterized previously¹³, which was used to guide the experiment in this study. As suggested, we clearly stated this in the main text.

Reference: 13 Wang, Y., Yan, Z. & Shan, X. Optimization of Process Parameters for Vertical-Faced Polypropylene Bottle Injection Molding. *Advances in Materials Science and Engineering* 2018, 2635084 (2018).
Please see the revised text as it appears on Page 12 Lines 509-510.

10. Line 39: Many drinking bottles are PET. Caps are often PP. PP is pretty brittle for water storage.

Response: We confirm that bottles made from both PET and PP are extensively used for water packaging and storage¹⁴. The primary objective of this study is to investigate the phase separation phenomenon in PP plastics, which represent the largest segment of modern plastic production, accounting for 21% of all non-fiber plastic production¹⁵. Accordingly, this research focused on PP bottles. It is important to note that further research is necessary to explore the phase separation phenomenon in PET and PET-based bottles.

Following your suggestions, we have revised the manuscript to enhance clarity.

References:

14 Chen, Y. et al. Plastic Bottles for Chilled Carbonated Beverages as a Source of Microplastics and Nanoplastics. *Water Research*, 120243 (2023).

15 Geyer, R., Jambeck, J. R. & Law, K. L. Production, use, and fate of all plastics ever made. *Science advances* 3, e1700782 (2017).

Please see the revised text as it appears on Page 2 Line 43-44.

11. Line 48: “no fundamental” is too broad. There is some understanding. Just not complete.

Response: As suggested, we revised the text to accurately identified the research gap in the Introduction.

Revised text: Despite the myriad of potential sources, most studies have focused on MP release mechanisms induced by extrinsic factors, such as UV irradiation and/or mechanical abrasion¹⁰⁻¹². Currently, there is no fundamental understanding of the intrinsic factors such as residual stress that are involved in the generation of MPs.

References:

10 Andrady, A. L. The plastic in microplastics: A review. *Marine Pollution Bulletin* 119, 12-22 (2017).

11 Shi, Y. et al. Formation of Nano-and Microplastics and Dissolved Chemicals During Photodegradation of Polyester Base Fabrics with Polyurethane Coating. *Environmental science & technology* 57, 1894-1906 (2023).

12 Song, Y. K. et al. Combined effects of UV exposure duration and mechanical abrasion on microplastic fragmentation by polymer type. *Environmental science & technology* 51, 4368-4376 (2017).

Please see the revised text as it appears on Page 2 Lines 52-55.

12. Line 56: define residual stress.

Response: We added the definition of residual stress as suggested. Residual stress refers to the locked-in stress distribution present in a structure, component, plate or sheet, while there are no external loads or forces applied².

Reference:

2 Schijve, J. *Fatigue of Structures and Materials*. 2nd edition, pp 89 (Springer Dordrecht, 2009).

Please see the revised text as it appears on Page 2 Lines 61-63.

13. Section starting line 59: Add more about the methods employed – e.g., if physical or computational in each case and map out the flow of the study to better introduce the reader to what you are doing and why.

Response: As suggested, we have added more detailed descriptions of the methods employed and the rationale for conducting this study, which has improved readability.

Added sentence: Here in this report, to unravel the impact of intrinsic residual stress on phase separation and subsequent MP release, we conducted cantilever beam experiments and a squeeze flow numerical study to demonstrate that residual stress drives nano- and microscale phase separation of the amorphous matrix onto the surfaces of plastics, from where it can be released as MPs.

Please see the revised text as it appears on Page 2 Lines 70-73.

14. Line 71: You “began” as the study was done in the past.

Response: We revised it to *began* as suggested.

Please see the revised text as it appears on Page 3 Line 84.

15. Line 73: How was vertical deflection attained?

Response: The vertical deflection was directly measured, as it shown at Fig. S2 (denoted as H) in Supplementary Information.

16. Line 120: note the glass transition temp for a-PP

Response: The glass transition temperature for a-PP ranges from -20 to -5 °C¹⁶. This value has been noted in the manuscript.

Reference: 16 Fred-Ahmadu, O. H. et al. Interaction of chemical contaminants with microplastics: Principles and perspectives. *Science of the Total Environment* 706, 135978 (2020).

Please see the revised text as it appears on Page 4 Lines 154.

17. Line 180: “the presence of the crystalline matrix through which the a-PP must pass through increases the barrier to a-PP flow” - unless you have direct evidence of this as the cause, then it is best to state it as a potential explanation.

Response: As suggested, we revised the sentence to highlight that this is a potential explanation.

Revised sentence: The larger barrier in our case is *likely* due to the crystalline matrix that impedes the flow.

Please see the revised text as it appears on Page 6 Lines 215-218.

18. Lines 173 to 182: what were the errors on your and the past estimate of the energy barrier to a-PP flow? Important to report the uncertainty and, in this case, to verify that your value of ~140 is significantly different to the previously reported value of 159.

Response: In the revised manuscript, we have included the error margin for the energy barrier to a-PP flow, now stated as 143.3 ± 4.3 kJ/mol. Regarding previously reported value, the error level was not available in the publication.

Please see the revised text as it appears on Page 5 Line 212.

19. Line 234: Present PE in as much detail as PP or explain why you do not do this in the main text.

Response: As suggested, we have incorporated detailed data and analysis of polyethylene (PE) into the main text. This addition substantially enhances the depth and scope of this paper. We summarised the results in the main text and detail the methods and experiment in the Supplementary Information.

Added text: In addition to PP plastic sheets, we investigated stress-driven amorphous phase separation on the surfaces of standard semicrystalline polyethylene (PE) sheets (Fig. R1, Below). By configuring a PE sheet into a cantilever setup (Table S1), we observed that only the upper compressive side of the cantilever, approximately 3 mm from the clamped end, developed a significant number of droplets. These droplets typically ranged from 10-40 μm in lateral size and 0.3-2 μm in height after being exposed to a 95 °C oven (Fig. R1e-g). In contrast, no observable changes occurred on the tensile surface or the compressive side near the free end, consistent with observations from PP sheets. Raman spectroscopy analysis indicated the absence of the peak at 1416 cm^{-1} , typically associated with methylene bending¹⁷. The degree of crystallinity in PE is also proportionally related to the intensity of this 1416 cm^{-1} peak (Fig. R1c)¹⁷. The absence of this specific band confirmed that the migrated droplets from the PE sheet were also amorphous. Following the same experimental protocol as with the PP sheet, the normalized volumetric flow (\hat{v}) of amorphous PE (a-PE) droplets was obtained as 26.6

$\text{nm}^3/(\text{nm}^2\cdot\text{h})$ or nm/h , which is higher than that observed for PP under the same compressive stress level (around -5 MPa, Fig. 2c), showing that the specific nature of polymer material influences the migration speed of amorphous polymer. Additionally, PE sheets formed into a cylindrical shape were also immersed in 95 °C DI water for 4 hours. Post-exposure inspection revealed a high quantity of circular shapes formed by air bubbles on the surface of the PE sheet (Fig. R1c). Residues of these partial circles were also observed, indicating dynamic processes of air bubble expansion and coalescence that substantially deformed these separated droplets. No a-PE droplets or circles were observed on the outer tensile surface of the cylindrical sheets. Although further research is required on other plastic types, the combined global market share of PP and PE, which accounts for 54%¹⁸, underscores the vital importance of stress-driven phase separation in the plastics industry.

References:

17 Strobl, G. & Hagedorn, W. Raman spectroscopic method for determining the crystallinity of polyethylene. *Journal of Polymer Science: Polymer Physics Edition* 16, 1181-1193 (1978).

18 Hahladakis, J. N., Velis, C. A., Weber, R., Iacovidou, E. & Purnell, P. An overview of chemical additives present in plastics: Migration, release, fate and environmental impact during their use, disposal and recycling. *Journal of hazardous materials* 344, 179-199 (2018).

Please see the revised text as it appears on Page 6 Lines 247-253, and Supplementary Information Page S3 and Page S18, Fig. S15.

Figure. R1 Analysis of surface stress driven a-PE phase separation and rearrangement in water environment. **a**, Optical image of virgin surface of standard PE sheet (low density). **b**, Optical image of PE compressive surface that close to clamped end after 95 °C oven heat. **c**, Optical image of inner compressed surface of PE sheet after 4hrs of 95 °C hot water exposure. **d**, Raman spectra of droplets from the surface of the stressed PE sheet, compared to the spectra from standard bulk PE sheet. **e-f**, AFM images of PE surface before and after 95 °C oven heat, both images have the same scale bars. **g**, The cross-section profiles of typical surface droplets in figure f.

Reviewer #2:

1. I find this research about the generation mechanisms of nano/microplastics at the surface of PP/PE fascinating and extremely timely and needed – the authors correctly state that we have a major global challenge, but we understand too little on the fundamental mechanisms.

First, I want to explicitly state that I am not a polymer materials scientist and therefore I am possibly not the best expert to assess the detail of the research here. The results are very noteworthy and intriguing. However, based on my wider expertise, I have some reservations on specific aspects and/or suggestions for improvement.

Response: Many thanks for your review. We appreciate your kind comments. As suggested, we have clearly answered the queries and comments point-by-point and revised the manuscript accordingly.

2. On the methodology: research includes a long series of experiments trying to quantify the response of the generation of formations on the surface of the plastic under a variety of conditions: Stress configuration settings (cantilever set up dimensions), temperature, time of exposure, other ambient conditions (nitrogen, vacuum and water) for both PE and PP. While the results are extensively reported in informative figures, what I miss is an over statistical design of experiments (DoE) that would allow a more wholistic testing and reporting of the effects here and in particular maybe provide insights on combined (confounded) effects. This is a standard practice. Maybe the tests have been conducted in a fashion that such a data analysis is also feasible?

Response: As suggested, experiments were designed and analysed using a full factorial methodology with four factors to comprehensively investigate the impact of various factors on phase separation and droplet emergence. Following established protocols, statistical analyses were conducted using ANOVA with a 95% confidence interval. Indeed, both surface stress and temperature significantly affect droplet emergence ($P < 0.05$). To quantitatively reveal the combined effects of surface stress and temperature on droplet flow, the Arrhenius model incorporating the parameter of temperature (T) was applied to conduct the analysis. By aggregating all data on a-PP flow across varying temperatures and stress levels, we were able to analyse the flow activation energy of a-PP, a critical parameter for understanding the nature of amorphous polymers¹⁹.

As suggested, we incorporated the experimental design and analysis into the manuscript.

Additionally, the confounded effect of surface stress and temperature was analysed using Arrhenius model.

Added text: To comprehensively investigate the impact of various factors on the emergence of droplets, experiments were designed and performed using a full factorial methodology with four factors (Fig. R2 and Table S2). ANOVA analysis reveals that phase separation and droplets formation is not dependent on the ambient environment but controlled by temperature and the local stress condition at the surface.

Added Analysis Method: To comprehensively investigate the impact of various factors on phase separation and droplet emergence, cantilever experiments were designed using a full factorial methodology with four factors: ambient condition (air, nitrogen and vacuum), surface stress (0 and 18 MPa), temperature (35, 60 and 95 °C) and exposure time (1 and 70 h), as shown in Figure S4. Following established protocols^{20,21}, statistical analyses were conducted using ANOVA with a 95% confidence interval (Minitab 22).

Figure Redacted

Figure. R2 The design of experiments using a full factorial methodology with four factors.

References:

19 Rosenzweig, N. & Narkis, M. Sintering rheology of amorphous polymers. *Polymer Engineering & Science* 21, 1167-1170 (1981).

20 Cambray, G., Guimaraes, J. C. & Arkin, A. P. Evaluation of 244,000 synthetic sequences reveals design principles to optimize translation in *Escherichia coli*. *Nature biotechnology* 36, 1005-1015 (2018).

21 Miloloža, M., Ukić, Š., Cvetnić, M., Bolanča, T. & Kučić Grgić, D. Optimization of polystyrene biodegradation by *Bacillus cereus* and *Pseudomonas alcaligenes* using full factorial design. *Polymers* 14, 4299 (2022).

Please see the revised text as it appears on Page 3 Lines 98-102, Page 9 Lines 336-340 and SI Fig.S4 and Table S2.

3. There are some extreme settings here, especially for the temperature range 35-95 °C. Not sure under which conditions the high range of temperature is of relevance here? Maybe in a desert setting? One thing is to try extreme temperatures to reveal the phenomena involved – and a different thing to establish relevance of plastic pollution problems. Especially, for example, when some of the effects are evident only at elevated temperatures: In Fig 2c: little /no effect is evident at 60 °C and below. Therefore, the relevance of the settings to nano/microplastics formation should be discussed. I might not have read carefully, but I do not fully understand the experimental setting on the temperature side. The standard sequence of stressing and then heating – does the sequence of applying heating then stressing been tested for? How this simulates the formation of the plastics, when I guess they are first heated and then stressed). How does this differentiation affect the testing relevance here?

Response: This study aims to elucidate the mechanisms of plastic degradation and microplastic release under environmental and daily-use conditions, which directly contribute to microplastic pollution and accumulation in environmental ecosystems and the human body. For accurate testing of environmental degradation of plastics, ISO standard 4892²² specifies methods simulating the weathering effects that occur when materials are exposed to actual environments. This standard mandates that the test temperature should be set between 50-70 °C to mimic real environmental conditions (e.g., midday in summer). During daily-use scenarios, the exposed temperature can easily reach 90-100 °C, as observed in plastic kettles, coffee cups and bottles. It should also be noted that PP is used almost exclusively for ‘keep cups’ for storing hot beverages. Additionally, EU Regulation 2020/1245²³ stipulates that plastic products experiencing any contact condition with temperatures exceeding 40 °C should be tested at these higher temperatures. Based on real conditions and

regulatory standards, we investigate the phase separation phenomenon under a temperature range of 35-95 °C. We have included this information in the paper to avoid any confusion.

In Figure 2C, while the a-PP migration at 60 °C is significant, it appears less pronounced when compared with the results at 95 °C, due to the higher Y-axis values at 95 °C overshadowing those at lower temperatures, rendering the 60 °C migration less visible. We confirm that the migration between 35-60 °C is clearly observed and accurately measured using AFM, as shown in Figure S3. The migration process between 35-60 °C took longer than at 95 °C, primarily because the viscosity and flow speed of a-PP are temperature-dependent. Regarding the test sequence, we initially set up the cantilever, then subjected it to different temperatures. This sequence typifies the lifecycle of plastic products, which accumulate residual stress during manufacturing and subsequently endure varying temperatures during daily use or after disposal in the environment. Indeed, a plastic product could first be heated and then mechanically stressed, for example, by stones if discarded in the environment. Above information has been included in the discussion as suggested.

References:

22 International Organization for Standardization. *Plastics—Methods of Exposure to Laboratory Light Sources ISO4892* (2016).

23 European Commission Regulation (EU) 2020/1245 of 2 September 2020 amending and correcting Regulation (EU) No 10/2011 on plastic materials and articles intended to come into contact with food. 1-17 (2020).

Please see the revised text as it appears on Page 6 Lines 225-227 and Page 7 Line 258.

4. At these elevated temperatures (95 °C), there might be other phenomena that could be taking place. Regarding the relevance to the wider plastic pollution what I miss from the discussion is how these nano- and micro plastics similar in any way to those documented in the literature from lab tests and in environmental/ field studies. Or there is nothing to compare with?

Response: Indeed, at elevated temperatures, additional phenomena may occur. For instance, small-molecular additives can co-migrate with amorphous PP polymer at these temperatures, increasing the volume of a-PP droplets on the plastic surface. To mitigate interference from additives, ethanol pretreatment was employed, as previously described²⁴. Regarding microplastics, amorphous-based microplastics have been found in the previous research (such as the MPs collected in the shorelines)^{25,26}. In line with suggestions, we have revised the Discussion section to compare our findings with prior reports and underscore the need for further research.

References:

24 Li, D. et al. Alcohol pretreatment to eliminate the interference of micro additive particles in the identification of microplastics using Raman spectroscopy. *Environmental science & technology* 56, 12158-12168 (2022).

25 Clunies-Ross, P., Smith, G., Gordon, K. & Gaw, S. Synthetic shorelines in New Zealand? Quantification and characterisation of microplastic pollution on Canterbury's coastlines. *New Zealand Journal of Marine and Freshwater Research* 50, 317-325 (2016).

26 Sorolla-Rosario, D., Llorca-Porcel, J., Pérez-Martínez, M., Lozano-Castello, D. & Bueno-Lopez, A. Study of microplastics with semicrystalline and amorphous structure identification by TGA and DSC. *Journal of Environmental Chemical Engineering* 10, 106886 (2022).

Please see the revised text as it appears on Page 3 Lines 115-116 and Page 8 Lines 314-316.

5. The authors the associate the formation of amorphous nano plastics on the surface of the polymer. Is there any physical explanation on the nature of the flow via the crystalline layers that sandwich the amorphous ones? Why this is predominantly an effect of stress and not elevated

temperature? This is what I personally conclude from the Figure 2 C – or at least that there is a step change in the behaviour (effect on normalised volumetric flow) from 85 to 95 °C: this is where a 3D plot of response surfaces may be helpful and DoE to assess and the comparative magnitude of the effect could help. So not fully convinced that the term “drives” is scientifically accurate in the title. This expression implies that this is the main mechanism and the variable that controls this mechanism, whilst it seems to me that temperature and phenomena associated with (driven by?) its change are clearly more potent here.

Response: We agree with your comment. Figure 2C demonstrates the synergistic effects of surface stress and ambient temperature on phase separation phenomena. From a physical perspective, the viscosity of pure amorphous polymers can decrease by an order of magnitude for every 20°C increase in temperature¹⁹. However, the presence of stress breaks the symmetry of the otherwise isotropic plastic and creates a direction of flow, that is absent without stress. It is in this sense that stress drive the direction of flow. Heating an unstressed PP sheet will not generate surface droplets. In our study, however, the presence of a crystalline matrix complicates the flow dynamics of amorphous PP (a-PP), introducing additional variables to consider. Despite these complexities, the significant role of elevated temperatures in facilitating a-PP flow is clear. To quantitatively assess the combined effects of surface stress and temperature on droplet flow, we employed the Arrhenius model, incorporating temperature as a critical parameter, to derive the critical flow parameters of a-PP, as detailed in our previous response.

As suggested, we revised the main text the highlight the influence of temperature on the flow. The synergistic effects were analysed using Arrhenius model.

Regarding the plot such as Figure 2C, 2D plot was used to benefit the subsequent data extraction (green line in Figure 2C) and flow energy barrier analysis (Figure 2e-f). This approach enhances readability and is consistent with commonly used methods in previous reports of fluid flow analysis^{27,28}. The use of a 2D plot thereby ensures that our data presentation aligns with established practices in the field. As suggested, we clearly stated this in Figure 2 and the main text.

References:

19 Rosenzweig, N. & Narkis, M. Sintering rheology of amorphous polymers. *Polymer Engineering & Science* 21, 1167-1170 (1981).

27 Vryzas, Z. et al. Effect of temperature on the rheological properties of neat aqueous Wyoming sodium bentonite dispersions. *Applied Clay Science* 136, 26-36 (2017).

28 Yanniotis, S., Skaltsi, S. & Karaburnioti, S. Effect of moisture content on the viscosity of honey at different temperatures. *Journal of Food Engineering* 72, 372-377 (2006).

Revised text: To describe the temperature dependent flow we note that the slope of each line in Fig. 2c is related to the viscosity of the a-PP matrix and differences in slope are consistent with the viscoplastic flow being activated and described by the Arrhenius equation.

Please see the revised text as it appears on Page 5 Lines 185-187, Lines 197-198 and Page 19 Figure 2 Lines 806-807.

6. There seems to be less controlled experimental settings with testes pertaining to release of the formulated nano-microplastics on the polymer surface into the water. I am not sure the phenomena around hot water of (95 °C) are accounted for and can only be associated with what it was tested in the atmospheric environment.

Response: The primary aim of this section is to understand stress-driven surface rearrangement and the subsequent release of nano- and microplastics in water. The experimental conditions were established in accordance with EU Regulation 2020/1245, which addresses scenarios in which plastic products encounter temperatures exceeding 40°C. We conducted a comprehensive analysis of four critical factors to deepen our understanding of stress-induced nano- and microplastic release in aquatic environments: stress levels (0, 2.5, 4, and 8 MPa), materials (PP and PE), product types (sheets and bottles) and product regions (bottle mouth and body). This investigation is vital, as the released nano- and microplastics are either consumed by users or discharged into waterways, posing threats to human and environmental health. However, for plastics discarded in environments subjected to lower temperatures but significant UV exposure, the dynamics of stress-driven phase separation and microplastic release may differ, which requires further study. As recommended, we have revised the main text to highlight the importance of the temperature setting.

Please see the revised text as it appears on Page 6 Lines 225-227.

7. The authors conclude: “Our findings underscore the need to modify current plastic processing technologies to reduce residual stress levels and suppress amorphous phase separation and microplastic release.” However, I am not sure how much of the potential for formulation of nano/microplastics could be mitigated by just washing the products in the first place? Also, a grey area remains establishing a better equivalence between the remaining stress in commercially produced plastics and the laboratory settings of the tests here. Maybe the authors could elaborate further in that.

Response: The pre-washing of products might be an effective method for reducing the release of nano- and microplastics while the choice of washing liquid, temperature and method are crucial. The most common wash liquids such as water and ethanol are not ideal for gentle soaking and rinsing due to the insolubility of amorphous PP polymers in these solvents, as demonstrated in Figures S7 and Figure 3b. A gentle scrub applying external force can effectively remove surface droplets, though some residual droplets may remain. Nevertheless, scrubbing may loosen particles, potentially facilitating the release of nano- and microplastics. We also tested chloroform and hexane, the typical laboratory solvents, which completely dissolved these droplets within two hours, consistent with the behavior of standard amorphous PP²⁹. Elevated temperatures should enhance wash speed and efficiency, given the increased solubility at higher temperatures. However, it is important to clarify that as long as the stress persists droplets will continue to be driven onto the plastic surface.

We fully recognize that establishing a precise equivalence between the residual stresses in commercially produced plastics and those in laboratory settings remains a complex challenge. While both the cantilever-applied stress in our tests and the residual stress from manufacturing can drive the migration of amorphous PP, our findings confirm that the cantilever can largely mimic these stresses. However, the real-world fabrication of plastic bottles introduces additional complexities that differ from the conditions used in our laboratory experiments. Unlike the flat and smooth surfaces of standard PP sheets used in labs, bottles with mouths and screw threads are formed using moulding techniques where mechanical clamping shapes the polymer under heat and pressure. This not only freezes polymer chains in specific directions but also generates high quantity of surface imperfections⁸, potentially altering the stress distribution and migration speed of amorphous PP. Further study requires to accurately identify these differences and develop products with low residual stress.

As suggested, we have included wash efficiency and stress differences in our discussion section.

Added text regarding wash: Ethanol may not effectively remove all oligomers or additives (there are 400 different types³⁰), while other common laboratory solvents like chloroform and hexane can completely dissolve the amorphous droplets within two hours due to the absence of protective crystalline regions found in the bulk semicrystalline plastics²⁹.

Regarding stress: The fabrication of plastic bottles also involves additional complexities. Bottles with mouths and screw threads are formed using moulding and blowing (where pre-formed shapes may increase in volume several times the original) techniques and mechanical clamps shape the polymer under heat and pressure. Often the blowing and former shape are critical in defining stress aligned crystallite directional growth to increase strength and barrier properties. These types of processes not only freeze polymer chains in specific directions but also generate large quantities of surface imperfections⁸, potentially altering the stress distribution and migration speed of a-PP^{1,9}, all of which is consistent with the enhanced MP release levels noted from the neck region of PP bottles.

References:

- 1 Guevara-Morales, A. & Figueroa-López, U. Residual stresses in injection molded products. *Journal of materials science* 49, 4399-4415 (2014).
- 8 Pedgley, O., Şener, B., Lilley, D. & Bridgens, B. Embracing material surface imperfections in product design. *International Journal of Design* (2018).
- 9 Syrett, D. Bottle design and manufacture and related packaging. *Carbonated Soft Drinks: Formulation and Manufacture*, 181-217 (2006).
- 29 Carraher Jr, C. E. & Seymour, R. *Structure—property relationships in polymers*. pp138 (Springer Science & Business Media, 2012).
- 30 European Chemicals Agency (ECHA). *Plastic Additives Initiative*, (2021), accessed December 07.

Please see the revised text as it appears on Page 8 Lines 284-287 and 295-303.

8. Despite these generic questions, my opinion is that the research reported here is very useful and critically advances our understanding. Maybe it could benefit from an overall more open discussion on its limitations, also referring to its relevance to actual average ambient conditions under which microplastics are generated.

Response: Many thanks for your kind comments. As suggested, we have added two paragraphs to extensively discuss the limitations of this study. These additions address the relevance of the test temperature range and potential differences between lab tests and real products. These revisions enhance the depth of the paper, providing substantial benefits to both readers and future research endeavors.

Main added text: Different responses are also found for the same polymer blended in different ways. PP sheets and PP bottles behave very differently, with the latter showing vastly higher levels of MP release. It is well known that a-PP is often directly incorporated into semicrystalline PP as a modifier to enhance the impact strength and thermal properties of final products such as bottles^{6,7}. This incorporation of a-PP not only increases the potential sources of a-PP migration but is known to result in reduced levels of polymer entanglement^{6,7}, which likely facilitates a-PP migration under stress conditions. The fabrication of plastic bottles also involves additional complexities. Bottles with mouths and screw threads are formed using moulding techniques where mechanical clamps shape the polymer under heat and pressure. This process not only freezes polymer chains in specific directions but also generates large quantities of surface imperfections⁸, potentially altering the stress distribution and migration speed of a-PP^{1,9}, all of which is consistent with the enhanced MP release levels noted from the neck region of PP bottles.

References:

- 1 Guevara-Morales, A. & Figueroa-López, U. Residual stresses in injection molded products. *Journal of materials science* 49, 4399-4415 (2014).
- 6 Nam, B.-K., Park, O. O. & Kim, S.-C. Properties of isotactic polypropylene/atactic polypropylene blends. *Macromolecular Research* 23, 809-813 (2015).
- 7 Chen, J.-H., Tsai, F.-C., Nien, Y.-H. & Yeh, P.-H. Isothermal crystallization of isotactic polypropylene blended with low molecular weight atactic polypropylene. Part I. Thermal properties and morphology development. *Polymer* 46, 5680-5688 (2005).
- 8 Pedgley, O., Şener, B., Lilley, D. & Bridgens, B. Embracing material surface imperfections in product design. *International Journal of Design* (2018).
- 9 Syrett, D. Bottle design and manufacture and related packaging. *Carbonated Soft Drinks: Formulation and Manufacture*, 181-217 (2006).

Please see the revised text as it appears on Page 7 Lines 289-303 and Page 6 Lines 225-227.

Reviewer #3:

1. The manuscript “Stress drives polymer phase separation and microplastic release into water” presents a study on the formation of microplastic on the surface of polypropylene (PP) and its release into water. The topic seems current and important, even though the problem of microplastics has existed for several decades without our awareness.

The performed tests are interesting; however, the presented interpretation of the data is fundamentally incorrect. According to the author’s interpretation, compressive stress and elevated temperature drive the nano- and microscale segregation of amorphous polymer droplets onto the surface of polypropylene. The analyzed polymer is a semicrystalline material. During solidification, its macromolecules are incorporated into the lamellar crystals. It is therefore not possible for macromolecules to be released from the crystals (without melting the crystals), disentangle from the molecular network of amorphous regions, and finally diffuse/migrate to the surface of the material, creating the mentioned domains, even at elevated temperature and under compressive stress. Even assuming that such migration of PP molecules takes place, why do they not crystallize within the droplets as demonstrated by spectroscopic techniques?

In my opinion, elevated temperature and compressive stress drive the migration of propylene oligomers and the formation of their droplets on the surface of the material. Initially, during the solidification process, the oligomers are expelled from growing crystals into intercrystalline regions. Due to the significantly smaller dimensions of the oligomer molecules compared to the PP molecules, their migration through amorphous regions to the material surface is more probable. This process is significantly accelerated at elevated temperatures. Moreover, the presence of compressive stresses further enhances the efficiency of oligomer migration to the material surface by decreasing the thickness of interlamellar regions (as schematically presented in Fig. S8c). Finally, conditioning PP samples at elevated temperatures enhances the degree of crystallinity of the polymer matrix. The reduced content of amorphous phase restricts the available “space” for oligomers, leading to their outflow from the interlamellar regions towards the material surface. It is also worth noting that the oligomers present in commercially available polypropylenes are chemically similar to polypropylene (PP) matrix or amorphous PP wax (in fact the atactic PP). Additionally, these oligomers practically do not crystallize (European Polymer Journal, 2015, 69, pp. 186–200, 6938), similar to the material found in the analyzed droplets.

Response: Many thanks for the construction suggestion. Following your advice, we have conducted additional experiments, yielding unequivocal data that confirm the polymer nature of these droplets. We have also carried out experiments to illustrate the diffusion process of amorphous PP within bulk semi-crystalline PP sheets. Further experiments verified that significant crystallization did not occur under the conditions explored in this study. Collectively, both our experimental results and existing literature validate the phenomena reported in this study. The detailed experimental data and analyses are summarized below.

(1) The confirmation of the polymer nature of the migrated droplets

We would disagree with the suggested mechanism as we have seen little evidence for the role of oligomers (i.e. with degrees of polymerization <40). Whilst the majority of longer chains will be associated with crystallite formation, shorter chains (note the $DPI > 3$) are likely present within the amorphous matrix. Gel Permeation Chromatography (GPC) equipped with an RI (Refractive Index) detector is a standard method for determining the molecular weight (MW) and molecular weight distribution (MWD) of polymers^{31,32}. In this study, GPC-RI was utilized to determine the MW and MWD of the separated droplets. The typical GPC curve displayed a primary peak alongside a smaller adjacent

peak, indicating two distinct molecular weight distributions³¹ (Fig. R3). The principal peak, accounting for 79% of the total mass of the droplets, corresponded to an average MW of 22,000 g/mol, which equates to a degree of polymerization (n) of 522. Notably, the MW exhibited a broad distribution range, with no single molecular weight constituting more than 50% of the weight of the substance (Cumulative fraction, Fig. R3b). The European Chemicals Agency (ECHA) defines a material as a polymer if it meets the following criteria³³: (1) > 50% of the weight of that substance consists of polymer molecules; and (2) the amount of polymer molecules presenting the same molecular weight must be < 50% of the weight of the substance. Combining spectroscopy and GPC results, it is evident that the separated droplets are polymer-based.

The average molecular weight (MW) associated with the minor peak, which accounts for 21% of the total mass of the droplets, is approximately 510 g/mol, corresponding to a degree of polymerization (n) of 12. Currently, information on the size cutoff between oligomers and polymers remains limited, although various regulations and research studies have used size cutoffs ranging from several to 40 repeat units³³⁻³⁵. Although migration of low molecular weight substances was not the predominant phenomenon observed in this study, the co-migration of oligomers and additives with polymers could be accelerated under harsh conditions, such as exposure to microwave heating or hot oil soaking. This indicates the necessity for further research to elucidate synergistic impacts of exposed conditions and surface stress on the migration behaviors of each component.

Figure R3: (a) The GPC detector response as a function of elution time for the separated droplets; (b) The molecular weight (MW) and molecular weight distribution (MWD) of main peak at figure (a).

(2) The confirmation of stress-driven diffusion/migration of amorphous PP

We conducted an experiment (refer to the following figure, R4) that confirmed the stress-driven migration of standard amorphous PP in a semi-crystalline PP bulk sheet. Standard amorphous PP (Goodfellow, average molecular weight 10,000 g/mol) was dissolved in chloroform and drop-casted on the surface of a semi-crystalline PP sheet to form standard amorphous droplets (SADs). These droplets closely resemble the separated droplets observed in our study. The area of the PP sheet under observation was subjected to a tensile stress of 4 MPa and heated to 95°C for 4 hours. Post-exposure, the surface reverted to its original smooth morphology with several manufacturing lines, and all SADs had migrated into the bulk of the semi-crystalline sheet. This is wholly consistent with a polymer above its glass transition temperature where higher chain mobility will drive a decrease in surface area. Subsequently, the same area was subjected to a compressive stress of -4 MPa and heated again to 95°C for another 4 hours. Post this second exposure, a large number of droplets accumulated in the

observed area (Fig. R4e). The volume of these emerged droplets was significantly larger than that of the original SADs or droplets that emerged on the surface of an unmodified plastic sheet under similar stress levels (Fig. 2a). This indicates that the compressive stress not only forced the incorporated standard amorphous PP to migrate but also drove the pre-existing amorphous PP within the sheet to the surface, resulting in a substantial increase in droplet flow. These observations clearly demonstrate that surface stress can effectively drive the migration of amorphous PP within a bulk semi-crystalline PP sheet.

Figure R4: The migration of standard amorphous PP droplets in semicrystalline PP sheet. **a**, Protocol used to confirm the migration of amorphous PP. A piece of semicrystalline PP sheet were thoroughly cleaned using DI water and observed using optical microscope. Standard amorphous PP (Goodfellow, PP306304/2, average molecular weight 10000 g/mol) was dissolved using chloroform and dropcasted onto the observed area on surface A, then allowed to air dry and form standard amorphous droplets. The PP sheet was then rolled into a cylindrical shape (diameter of 30 mm, surface A facing outward) using a stainless-steel O-ring. A 4 MPa tensile stress was applied to the observed area using this setting. The PP sheet was then exposed to 95°C for 4 hrs and the observed area was subsequently imaged. After that, the rolling direction of PP sheet was reversed to apply compressive stress (-4 MPa) to the observed area (diameter of 30 mm, surface A facing outward). The cylindrical PP sheet was exposed to 95 °C for another 4 hrs, and the observed area was then imaged again. **b**, Raw surface of the semicrystalline PP sheet. **c**, Observed surface after drop-casting and air-drying of amorphous PP. **d**, Observed surface after 4 hours at 95°C with 4 MPa tensile stress. **e**, Observed surface after 4 hours at 95°C with -4 MPa compressive stress. Figures **b-e** share the same scale bar. They are not from the exact same locations but are randomly selected representative images from an observed area of approximately 1 cm x 1 cm.

(3) The confirmation of no significant crystallization of amorphous PP under the experimental conditions.

X-ray diffraction (XRD) analyses were performed on standard amorphous polypropylene (a-PP) wax (Figure R5). The initial crystallinity of the raw a-PP wax was approximately 2.0%, attributed primarily to the presence of a diffraction peak at around 17° (2θ). This observation aligns with previous studies indicating that a-PP can contain a small fraction of crystalline structures, generally less than 5%³⁶. The sample was then subjected to a controlled thermal treatment, maintained at 95 °C for four hours—a typical condition for this study. Subsequent XRD analysis, conducted after the sample had cooled, revealed a slight decrease in crystallinity to 1.6%. These results suggest that significant crystallization did not occur under the applied test conditions.

Figure R5: XRD pattern of standard PP wax (a) before and (b) after 4 hrs 95 °C oven heat, respectively. Each XRD image was deconvoluted into two broad amorphous peaks and one sharp crystalline peak. The degree of crystallinity was obtained based on the area proportion of the crystalline peak relative to the total area.

Taken together, the experimental data presented above and relevant previous publications unequivocally confirm the phase separation phenomenon. As suggested, we have incorporated relevant figures, analyses and discussions into the paper to significantly enhance both the clarity and quality of this study.

Main added text: Gel Permeation Chromatography (GPC) was used to determine the molecular weight (MW) and molecular weight distribution (MWD) of the original PP plastic sheet and the separated droplets. The sheet has an average MW of 97,000 g/mol and a polydispersity index (PDI) of 3.6 (see Fig S8), consistent with a wide distribution of polymer chain lengths. The droplet GPC curve showed a main peak and a smaller peak, indicating two distinct molecular weight distributions (Fig. 1d). The main peak, accounted for 79% of the total mass of the droplets, corresponded to an average MW of 22,000 g/mol (degree of polymerization $n = 522$), which is significantly less than the average MW of the original sheet. The minor peak had an average MW of 510 g/mol. The main GPC peak is evidently due to the presence of low MW polymers in the origin PP sheet that either comprised the amorphous matrix or were generated under flow conditions by peeling off weakly bound short polymer chains from crystalline domains due to their reduced van der Waals interactions³⁷. The minor peak is likely due to chemical additives that are insoluble in alcohol³⁸ or possibly oligomers with $n \approx 12$. We note that oligomer release has been reported for plastics in contact with oils and fats during microwave exposure³⁸. These data unequivocally demonstrate a significant mobility of short chain polymer and molecular species in plastics under modest stress conditions. To further confirm this we performed a series of experiments in which amorphous PP droplets (Goodfellow, average MW 10,000 g/mol) were intentionally deposited onto the same standard semicrystalline PP sheet (49.7% crystallinity). The droplets were found to easily move from the surface into the bulk sheet under tensile stress and then to re-emerge onto the surface following the application of a compressive stress (Fig. S9). This facile movement of low MW polymer chains is responsible for droplet formation and expected to be a general feature of typical semicrystalline plastics with high levels of polydispersity (see below).

References:

- 31 Wang, J., Tsou, A. H. & Favis, B. D. Effects of polyethylene molecular weight distribution on phase morphology development in poly (p-phenylene ether) and polyethylene blends. *Macromolecules* 51, 9165-9176 (2018).
- 32 Kaneko, T., Thi, T. H., Shi, D. J. & Akashi, M. Environmentally degradable, high-performance thermoplastics from phenolic phytomonomers. *Nature materials* 5, 966-970 (2006).
- 33 European Chemicals Agency (ECHA). How to decide whether a substance is a polymer or not and how to proceed with the relevant registration. (2017.12).
- 34 Yang, T., Xu, Y., Liu, G. & Nowack, B. Oligomers are a major fraction of the released submicrometre particles released during washing of polyester textiles. *Nature Water*, 1-10 (2024).

35 International Organization for Standardization, Biological evaluation of medical devices. *Biol. Eval. Med. Devices* 1, 10993 (2003).

36 Wilkinson, R. W. & Dole, M. Specific heat of synthetic high polymers. X. Isotactic and atactic polypropylene. *Journal of Polymer Science* 58, 1089-1106 (1962).

37 Bartczak, Z. Deformation of semicrystalline polymers—the contribution of crystalline and amorphous phases. *Polimery* 62, 787-799 (2017).

38 Conchione, C., Lucci, P. & Moret, S. Migration of polypropylene oligomers into ready-to-eat vegetable soups. *Foods* 9, 1365 (2020).

Please see the revised text as it appears on Page 3 Lines 117-138, Page 9 Lines 373-385, Page 18 Fig 1d-e, and SI Fig.S8-9 and Page S2 Text S1.

2. To sum up, this manuscript observes and analyzes the process of oligomer migration, which is an issue addressed in other works (e.g., *Foods* 2020, 9(10), 1365), rather than focusing on the formation and release of microplastics.

Response: As previously discussed, we have provided conclusive data confirming the polymer-based nature of the migrated droplets observed in our study. Thank you for the suggested paper to compare with our findings. The suggested paper³⁸ (*Foods* 2020, 9(10), 1365) primarily focuses on the effects of microwave irradiation on the migration of oligomers (C10-35 alkane) into food during typical daily-use conditions. It is well-documented that microwave irradiation can degrade polyolefins such as PP and PE due to cavitation effects^{39,40}. Specifically, a study on PP highlighted that just 90 seconds of microwave irradiation in the presence of potassium permanganate induced significant surface oxidation, an effect not seen with potassium permanganate alone⁴¹. Further research is necessary to verify the generation of oligomers induced by microwave irradiation and assess their potential contribution to oligomer migration as discussed in the paper³⁸ (*Foods* 2020, 9(10), 1365). However, it is clear that the mechanisms described in that paper, focusing on microwave-induced changes, differ substantially from the stress-driven migration outlined in our study.

Additionally, we investigated the thermal stability to underscore the significant differences between the separated droplets in our study and the oligomers (C10-35 alkane) reported in³⁸ (*Foods* 2020, 9(10), 1365). It is well-established that the melting point of oligomer alkanes increases with the carbon chain length⁴². For instance, C35 (Pentatriacontane) has a melting temperature of 74.5 °C, indicating that the melting points of all observed oligomers³⁸ (C10-35 alkane) are below 74.5 °C. When we exposed the migrated droplets in our study to 80 °C water for 10 minutes, the majority remained stable (Figure R6). Furthermore, most of these droplets maintained their initial shape and size even after 4 hours of exposure to 95 °C water, provided no air bubbles formed in the observed area (Figure 3b, main text). Evidently, the amorphous droplets observed in our study are distinct from the C10-35 alkane types reported previously³⁸, but instead are predominantly polymers with an average MW of 22,000 g/mol (Fig R3 above).

As suggested, we have revised the Discussion to emphasize the importance of further investigating potential migration induced by surface stress.

Added text: We note that oligomer release has been reported for plastics in contact with oils and fats during microwave exposure³⁸. These data unequivocally demonstrate a significant mobility of short chain polymer and molecular species in plastics under modest stress conditions.

Figure R6: In-situ AFM imaging of a PP sheet surface. (a) Displays the surface with separated a-PP droplets and (b) Shows the surface after a 10-minute soak in 80°C water. The PP sheet was initially stressed and heated in a 95 °C oven to facilitate the formation of a-PP droplets. Post-heating, the PP sheet was imaged using AFM. Following this, the sheet was soaked in 80°C water, then removed, dried with N₂ gas, and reimaged at the same location using AFM for comparison.

References:

- 38 Conchione, C., Lucci, P. & Moret, S. Migration of polypropylene oligomers into ready-to-eat vegetable soups. *Foods* 9, 1365 (2020).
- 39 Bäckström, E., Odelius, K. & Hakkarainen, M. Trash to treasure: microwave-assisted conversion of polyethylene to functional chemicals. *Industrial & Engineering Chemistry Research* 56, 14814-14821 (2017).
- 40 Achilias, D. S. Polymer degradation under microwave irradiation. *Microwave-assisted Polymer Synthesis*, 309-346 (2016).
- 41 Mallakpour, S. E., Hajipour, A. R., Zadhoush, A. & Mahdavian, A. R. Efficient and novel method for surface oxidation of polypropylene in the solid phase using microwave irradiation. *Journal of applied polymer science* 79, 1317-1323 (2001).
- 42 Chickos, J. S. & Hanshaw, W. Vapor Pressures and Vaporization Enthalpies of the n-Alkanes from C31 to C38 at T= 298.15 K by Correlation Gas Chromatography. *Journal of Chemical & Engineering Data* 49, 620-630 (2004).

Please see the revised text as it appears on Page 4 Lines 129-132.

3. Other remarks. No ethanol is used to remove non-polymeric substances (additives, oligomers) from polyolefins (PP or PE). Typically, a mixture of hexane, chloroform, and ethanol (*Polym. Degrad. Stab.* 1998, 60, 137–143; *Polym. Degrad. Stab.* 2006, 91, 1598–1605) is used for effective removal of such substances.

Response: Many thanks for the suggestion. Although solvent mixture have been employed with bulk semi-crystalline plastics, they are not suitable for removing non-polymeric substances (such as additives and oligomers) from amorphous polypropylene. Previous research has indicated that amorphous polypropylene is soluble in hexane due to the absence of protective crystalline structures typically found in semi-crystalline plastics²⁹. Furthermore, the use of a hexane-based method to extract amorphous polyolefins has been patented⁴³.

As suggested, we conducted relevant experiments to confirm that amorphous polypropylene is soluble in mixed solvents, as well as in hexane and chloroform separately. For instance, we prepared a mixture of hexane, chloroform and ethanol in a ratio of 4:1:1, following the methodologies described in previous studies^{44,45} (*Polym. Degrad. Stab.* 1998, 60, 137–143; *Polym. Degrad. Stab.* 2006, 91, 1598–1605). After soaking for 2 hours at room temperature, 0.5 g of standard amorphous polypropylene was completely dissolved in 50 mL of the mixed solvent (Figure R7), aligning with findings reported in previous research²⁹. When tested individually, both hexane and chloroform were also effective in dissolving amorphous polypropylene. In contrast, ethanol did not dissolve the amorphous polypropylene. Additionally, previous studies have demonstrated ethanol's high efficiency in removing

additives²⁴ and oligomers³⁴ from polymer materials. Consequently, in this study, ethanol was specifically used to remove additives and oligomers while preserving the integrity of the amorphous polymer. We fully acknowledge the reviewer's concern that ethanol may not completely remove all oligomers or additives, considering the extensive use of over 400 different types of additives in plastics³⁰.

As suggested, we have incorporated the relevant references and revised the Methods section to clearly articulate the rationale for using ethanol as a solvent. Additionally, we have discussed the potential limitations of each solvent and highlighted the need for further study.

Figure R7: Dissolution of standard amorphous polypropylene in mixture solvent of hexane, chloroform and ethanol in a ratio of 4:1:1. **a**, 0.5 g standard amorphous PP was projected on a glass slide and fully submerged in the mixed solvent. **b**, the same spot after 2-hour soak at room temperature (approximately 20 °C). Figures **a-b** share the same scale bar. The complete dissolution of the standard amorphous PP is evident, as indicated by the change in color of the mixture solvent.

References:

- 24 Li, D. et al. Alcohol pretreatment to eliminate the interference of micro additive particles in the identification of microplastics using Raman spectroscopy. *Environmental science & technology* 56, 12158-12168 (2022).
- 29 Carraher Jr, C. E. & Seymour, R. *Structure—property relationships in polymers*. pp138 (Springer Science & Business Media, 2012).
- 30 European Chemicals Agency (ECHA). Plastic Additives Initiative, (2021), accessed December 07.
- 34 Yang, T., Xu, Y., Liu, G. & Nowack, B. Oligomers are a major fraction of the released submicrometre particles released during washing of polyester textiles. *Nature Water*, 1-10 (2024).
- 43 Robinson, J. & Folzenlogen, P. (Google Patents, 1974).
- 44 Khelidj, N. et al. Oxidation of polyethylene under irradiation at low temperature and low dose rate. Part II. Low temperature thermal oxidation. *Polymer degradation and stability* 91, 1598-1605 (2006).
- 45 Audouin, L., Girois, S., Achimsky, L. & Verdu, J. Effect of temperature on the photooxidation of polypropylene films. *Polymer degradation and stability* 60, 137-143 (1998).

Please see the revised text as it appears on Page 7 Lines 284-287.

4. There is a lack of basic information about the materials analyzed in the work. How was the degree of crystallinity determined?

Response: To determine the degree of crystallinity, both X-ray diffraction (XRD) and Raman spectroscopy were performed on standard semi-crystalline PP sheets, standard amorphous PP and PP

bottles. To confirm the amorphous nature of the separated droplets, Raman spectroscopy was conducted and the Hit Quality Index (HQI) was analysed, following established protocols^{46,47}. Due to the high mass requirement for XRD tests (typically 1 gram), this method was not applied to the nano- and micro-sized droplets.

Overall, this study comprehensively characterized the physicochemical properties of the materials using a range of techniques, including Raman spectroscopy, XRD, atomic force microscopy (AFM), scanning electron microscopy (SEM), Fourier-transform infrared spectroscopy (FTIR) and gel permeation chromatography (GPC). We are confident that the data collected are sufficient to confirm the observed phenomena.

As suggested, we have incorporated both the XRD and Raman spectroscopy test methods and results into the paper.

Please see the revised text as it appears on Page 3 Line 85, Page 9 Lines 373-376, SI fig. S1 and SI Page S2 Text 1.

Added content:

Method: X-ray diffraction (XRD) measurements were performed using a diffractometer (Bruker D8 Discover) equipped with Cu K α 1 radiation. Data were collected at room temperature over a scan range of 5° to 35°. After background subtraction, the XRD pattern was delineated into crystalline peaks (total area, A_c) and an amorphous halo (A_a), following the previously established protocol⁴⁸. The degree of crystallinity was calculated using Equation R1 (Figure R8)

$$x_c^{XRD} = \frac{A_c}{A_a + A_c} \quad (R1)$$

Raman spectroscopy was also conducted to obtain Raman spectra and the crystallinity property of samples. The spectra were obtained in the range of 250-3500 cm⁻¹. WiRE 3.4 (Renishaw) software was used to operate the system and analyze the recorded spectra. After detection, the hit quality index (HQI) analysis comparing with standard PP was conducted. A higher value of HQI indicates the greater match between the sample and known materials, facilitating accurate identification of chemical property.

Results:

XRD analysis revealed that the crystallinity degree of the standard PP sheet was 49.7°. The Raman spectra distinctly showed the crystalline C-C backbone vibration at 1168 cm⁻¹. Additionally, two sharp peaks at 810 and 840 cm⁻¹, associated with helical chains within the crystals, were observed, aligning with previous reports^{46,49}. Combined with the XRD results, it is evident that the standard PP sheet is semi-crystalline, containing both crystalline and amorphous regions.

Interestingly, distinct differences were observed between the PP droplets and the parent PP sheet in the low wavenumber range (800-1500 cm⁻¹), particularly at 810 and 840 cm⁻¹, which are associated with the crystallinity degree of PP. Additionally, the PP droplets lacked the peak at 1168 cm⁻¹, which is associated with the crystalline C-C backbone vibration^{46,49} (Fig. R8b). These differences resulted in a poor match between the PP droplets and the parent PP sheet, with a Hit Quality Index (HQI) of only 0.75. In contrast, the droplets closely matched the spectrum of standard amorphous PP wax (Goodfellow), achieving an HQI of 0.93 in the low wavenumber range (800-1500 cm⁻¹, Fig. R8b).

Figure R8: (a) XRD pattern of the standard semicrystalline PP sheet; (b) Raman spectra of bulk PP sheet, a-PP droplets from the surface of standard PP sheet and standard amorphous PP wax, respectively.

References:

- 46 Nielsen, A. S., Batchelder, D. & Pyrz, R. Estimation of crystallinity of isotactic polypropylene using Raman spectroscopy. *Polymer* 43, 2671-2676 (2002).
- 47 Levermore, J. M., Smith, T. E., Kelly, F. J. & Wright, S. L. Detection of microplastics in ambient particulate matter using Raman spectral imaging and chemometric analysis. *Analytical Chemistry* 92, 8732-8740 (2020).
- 48 Machado, G. et al. Crystalline properties and morphological changes in plastically deformed isotactic polypropylene evaluated by X-ray diffraction and transmission electron microscopy. *European Polymer Journal* 41, 129-138 (2005).
- 49 Gopanna, A., Mandapati, R. N., Thomas, S. P., Rajan, K. & Chavali, M. Fourier transform infrared spectroscopy (FTIR), Raman spectroscopy and wide-angle X-ray scattering (WAXS) of polypropylene (PP)/cyclic olefin copolymer (COC) blends for qualitative and quantitative analysis. *Polymer Bulletin* 76, 4259-4274 (2019).

5. I do not recommend the manuscript for publication because of the incorrect interpretation of experimental data and the lack of novelty.

Response: Many thanks for your constructive suggestions. We have conducted additional and extensive studies to verify the accuracy of our findings. Following your recommendations, we have revised the discussion section of our draft to clearly delineate the differences between this study and the results reported in the previous publication³⁸ (*Foods* 2020, 9(10), 1365). Our research provides fundamental new insights into the microplastics release mechanism in water environments. We believe that these detailed responses address your concerns comprehensively.

Reference:

- 38 Conchione, C., Lucci, P. & Moret, S. Migration of polypropylene oligomers into ready-to-eat vegetable soups. *Foods* 9, 1365 (2020).

Full Reference:

- 1 Guevara-Morales, A. & Figueroa-López, U. Residual stresses in injection molded products. *Journal of materials science* 49, 4399-4415 (2014).
- 2 Schijve, J. *Fatigue of Structures and Materials*. 2 edn, pp 89 (Springer Dordrecht, 2009).
- 3 Spina, R., Spekowiak, M., Dahlmann, R. & Hopmann, C. Analysis of polymer crystallization and residual stresses in injection molded parts. *International journal of precision engineering and manufacturing* 15, 89-96 (2014).

- 4 Mills, N. Residual stresses in plastics, rapidly cooled from the melt, and their relief by sectioning. *Journal of Materials Science* **17**, 558-574 (1982).
- 5 Cho, S.-H., Hong, J.-S. & Lyu, M.-Y. Investigation of the molding conditions to minimize residual stress and shrinkage in injection molded preform of PET bottle. *Polymer (Korea)* **35**, 467-471 (2011).
- 6 Nam, B.-K., Park, O. O. & Kim, S.-C. Properties of isotactic polypropylene/atactic polypropylene blends. *Macromolecular Research* **23**, 809-813 (2015).
- 7 Chen, J.-H., Tsai, F.-C., Nien, Y.-H. & Yeh, P.-H. Isothermal crystallization of isotactic polypropylene blended with low molecular weight atactic polypropylene. Part I. Thermal properties and morphology development. *Polymer* **46**, 5680-5688 (2005).
- 8 Pedgley, O., Şener, B., Lilley, D. & Bridgens, B. Embracing material surface imperfections in product design. *International Journal of Design* (2018).
- 9 Syrett, D. Bottle design and manufacture and related packaging. *Carbonated Soft Drinks: Formulation and Manufacture*, 181-217 (2006).
- 10 Andrady, A. L. The plastic in microplastics: A review. *Marine Pollution Bulletin* **119**, 12-22 (2017).
- 11 Shi, Y. *et al.* Formation of Nano-and Microplastics and Dissolved Chemicals During Photodegradation of Polyester Base Fabrics with Polyurethane Coating. *Environmental science & technology* **57**, 1894-1906 (2023).
- 12 Song, Y. K. *et al.* Combined effects of UV exposure duration and mechanical abrasion on microplastic fragmentation by polymer type. *Environmental science & technology* **51**, 4368-4376 (2017).
- 13 Wang, Y., Yan, Z. & Shan, X. Optimization of Process Parameters for Vertical-Faced Polypropylene Bottle Injection Molding. *Advances in Materials Science and Engineering* **2018**, 2635084 (2018).
- 14 Chen, Y. *et al.* Plastic Bottles for Chilled Carbonated Beverages as a Source of Microplastics and Nanoplastics. *Water Research*, 120243 (2023).
- 15 Geyer, R., Jambeck, J. R. & Law, K. L. Production, use, and fate of all plastics ever made. *Science advances* **3**, e1700782 (2017).
- 16 Fred-Ahmadu, O. H. *et al.* Interaction of chemical contaminants with microplastics: Principles and perspectives. *Science of the Total Environment* **706**, 135978 (2020).
- 17 Strobl, G. & Hagedorn, W. Raman spectroscopic method for determining the crystallinity of polyethylene. *Journal of Polymer Science: Polymer Physics Edition* **16**, 1181-1193 (1978).
- 18 Hahladakis, J. N., Velis, C. A., Weber, R., Iacovidou, E. & Purnell, P. An overview of chemical additives present in plastics: Migration, release, fate and environmental impact during their use, disposal and recycling. *Journal of hazardous materials* **344**, 179-199 (2018).
- 19 Rosenzweig, N. & Narkis, M. Sintering rheology of amorphous polymers. *Polymer Engineering & Science* **21**, 1167-1170 (1981).
- 20 Cambray, G., Guimaraes, J. C. & Arkin, A. P. Evaluation of 244,000 synthetic sequences reveals design principles to optimize translation in *Escherichia coli*. *Nature biotechnology* **36**, 1005-1015 (2018).
- 21 Miloloža, M., Ukić, Š., Cvetnić, M., Bolanča, T. & Kučić Grgić, D. Optimization of polystyrene biodegradation by *Bacillus cereus* and *Pseudomonas alcaligenes* using full factorial design. *Polymers* **14**, 4299 (2022).
- 22 International Organization for Standardization. Plastics—Methods of Exposure to Laboratory Light Sources ISO4892 (2016).
- 23 European Commission Regulation (EU) 2020/1245 of 2 September 2020 amending and correcting Regulation (EU) No 10/2011 on plastic materials and articles intended to come into contact with food. 1-17 (2020).
- 24 Li, D. *et al.* Alcohol pretreatment to eliminate the interference of micro additive particles in the identification of microplastics using Raman spectroscopy. *Environmental science & technology* **56**, 12158-12168 (2022).
- 25 Clunies-Ross, P., Smith, G., Gordon, K. & Gaw, S. Synthetic shorelines in New Zealand? Quantification and characterisation of microplastic pollution on Canterbury's coastlines. *New Zealand Journal of Marine and Freshwater Research* **50**, 317-325 (2016).
- 26 Sorolla-Rosario, D., Llorca-Porcel, J., Pérez-Martínez, M., Lozano-Castello, D. & Bueno-Lopez, A. Study of microplastics with semicrystalline and amorphous structure identification by TGA and DSC. *Journal of Environmental Chemical Engineering* **10**, 106886 (2022).
- 27 Vryzas, Z. *et al.* Effect of temperature on the rheological properties of neat aqueous Wyoming sodium bentonite dispersions. *Applied Clay Science* **136**, 26-36 (2017).

- 28 Yanniotis, S., Skaltsi, S. & Karaburnioti, S. Effect of moisture content on the viscosity of honey at different temperatures. *Journal of Food Engineering* **72**, 372-377 (2006).
- 29 Carraher Jr, C. E. & Seymour, R. *Structure—property relationships in polymers*. pp138 (Springer Science & Business Media, 2012).
- 30 European Chemicals Agency (ECHA). *Plastic Additives Initiative*, (2021), accessed December 07.
- 31 Wang, J., Tsou, A. H. & Favis, B. D. Effects of polyethylene molecular weight distribution on phase morphology development in poly (p-phenylene ether) and polyethylene blends. *Macromolecules* **51**, 9165-9176 (2018).
- 32 Kaneko, T., Thi, T. H., Shi, D. J. & Akashi, M. Environmentally degradable, high-performance thermoplastics from phenolic phytomonomers. *Nature materials* **5**, 966-970 (2006).
- 33 European Chemicals Agency (ECHA). How to decide whether a substance is a polymer or not and how to proceed with the relevant registration. (2017.12).
- 34 Yang, T., Xu, Y., Liu, G. & Nowack, B. Oligomers are a major fraction of the released submicrometre particles released during washing of polyester textiles. *Nature Water*, 1-10 (2024).
- 35 International Organization for Standardization, Biological evaluation of medical devices. *Biol. Eval. Med. Devices* **1**, 10993 (2003).
- 36 Wilkinson, R. W. & Dole, M. Specific heat of synthetic high polymers. X. Isotactic and atactic polypropylene. *Journal of Polymer Science* **58**, 1089-1106 (1962).
- 37 Bartczak, Z. Deformation of semicrystalline polymers—the contribution of crystalline and amorphous phases. *Polimery* **62**, 787-799 (2017).
- 38 Conchione, C., Lucci, P. & Moret, S. Migration of polypropylene oligomers into ready-to-eat vegetable soups. *Foods* **9**, 1365 (2020).
- 39 Bäckström, E., Odelius, K. & Hakkarainen, M. Trash to treasure: microwave-assisted conversion of polyethylene to functional chemicals. *Industrial & Engineering Chemistry Research* **56**, 14814-14821 (2017).
- 40 Achilias, D. S. Polymer degradation under microwave irradiation. *Microwave-assisted Polymer Synthesis*, 309-346 (2016).
- 41 Mallakpour, S. E., Hajipour, A. R., Zadhoush, A. & Mahdavian, A. R. Efficient and novel method for surface oxidation of polypropylene in the solid phase using microwave irradiation. *Journal of applied polymer science* **79**, 1317-1323 (2001).
- 42 Chickos, J. S. & Hanshaw, W. Vapor Pressures and Vaporization Enthalpies of the n-Alkanes from C31 to C38 at T= 298.15 K by Correlation Gas Chromatography. *Journal of Chemical & Engineering Data* **49**, 620-630 (2004).
- 43 Robinson, J. & Folzenlogen, P. (Google Patents, 1974).
- 44 Khelidj, N. *et al.* Oxidation of polyethylene under irradiation at low temperature and low dose rate. Part II. Low temperature thermal oxidation. *Polymer degradation and stability* **91**, 1598-1605 (2006).
- 45 Audouin, L., Girois, S., Achimsky, L. & Verdu, J. Effect of temperature on the photooxidation of polypropylene films. *Polymer degradation and stability* **60**, 137-143 (1998).
- 46 Nielsen, A. S., Batchelder, D. & Pyrz, R. Estimation of crystallinity of isotactic polypropylene using Raman spectroscopy. *Polymer* **43**, 2671-2676 (2002).
- 47 Levermore, J. M., Smith, T. E., Kelly, F. J. & Wright, S. L. Detection of microplastics in ambient particulate matter using Raman spectral imaging and chemometric analysis. *Analytical Chemistry* **92**, 8732-8740 (2020).
- 48 Machado, G. *et al.* Crystalline properties and morphological changes in plastically deformed isotactic polypropylene evaluated by X-ray diffraction and transmission electron microscopy. *European Polymer Journal* **41**, 129-138 (2005).
- 49 Gopanna, A., Mandapati, R. N., Thomas, S. P., Rajan, K. & Chavali, M. Fourier transform infrared spectroscopy (FTIR), Raman spectroscopy and wide-angle X-ray scattering (WAXS) of polypropylene (PP)/cyclic olefin copolymer (COC) blends for qualitative and quantitative analysis. *Polymer Bulletin* **76**, 4259-4274 (2019).

Responses to the referees

We would like to thank the editor and the reviewers for reviewing our manuscript again. We agree with most of the comments and the manuscript has been revised accordingly. All changes are highlighted in blue and can be found in the word document (main text-blue tracked.doc). Our responses to the specific comments of the editor and reviewers are summarized below.

Reviewer #3:

1. This is a review of the work titled: Stress Drives Polymer Phase Separation and Microplastic Release into Water.

The test performed using the GPC technique clearly demonstrated that the molecular weight (MW) of the main fraction of substances in the droplets is five times lower than the average MW of the analyzed polypropylene. Additionally, the lack of crystallization ability is a manifestation of insufficient regioregularity (isotacticity) of the chains, similar to atactic polypropylene wax. Therefore, from both a physical and chemical perspective, the material in the droplets has completely different properties compared to the polypropylene analyzed in the manuscript. Due to the lower MW, lack of crystallinity, and regioregularity of the chains, the glass transition temperature (T_g) of this amorphous mixture is likely shifted 20-30°C lower compared to the T_g of polypropylene. Thus, at room temperature, the material within the droplets exhibits a waxy consistency, akin to atactic polypropylene wax.

According to the generally accepted definitions (U.S. National Oceanic and Atmospheric Administration, U.S. Environmental Protection Agency, or European Chemicals Agency), microplastics are small plastic pieces or particles less than five millimeters in size.... The term “pieces or particles” refers to objects that are solid under given (environmental) conditions. Therefore, the use of the term “microplastic” in relation to droplets of substances (with inhomogeneous content) that have a gooey or waxy consistency seems unjustified.

Response: Thank you again for the insightful comments. We appreciate your detailed analysis of the properties of the semi-solid droplets compared to the solid bulk polypropylene. We fully acknowledge that the physical and perhaps the chemical properties of semi-solid droplets are different from that of the bulk polypropylene, as you have pointed out. Indeed, the droplets exhibit lower molecular weight, reduced crystallinity, and differences in regioregularity, leading to a waxy consistency at room temperature.

Regarding the terminology, we agree that discussions of microplastics in the literature typically envision solid particles. However, according to the definitions provided by regulatory agencies such as the European Chemicals Agency (ECHA) [1, pp29-30], semi-solid particles are included within the scope of microplastics.

Clarification on the Definition of Microplastics:

ECHA Definition: In March 2018, ECHA proposed defining microplastics as any polymer, or polymer-containing, **solid or semi-solid** particle having a size of 5mm or less in at least one external dimension [1, pp29]. In August 2019, ECHA further detailed the definition of microplastic as a material consisting of **solid** polymer-containing particles [1, pp30]. They clarified that '**solid**' refers to a substance or mixture that does not meet the definitions of a liquid or gas [1, pp30]. Specifically:

(1) **gas** means a substance which (i) at 50 °C has a vapor pressure greater than 300 kPa (absolute), or (ii) is completely gaseous at 20 °C and a standard pressure of 101.3 kPa.

(2) **liquid** is a substance or mixture which (i) at 50 °C has a vapor pressure not more than 300 kPa, (ii) is not completely gaseous at 20 °C and standard pressure of 101.3 kPa, and (iii) has a melting point or initial melting point of 20 °C or less at standard pressure.

Classification of a-PP Wax: The atactic polypropylene (a-PP) wax in semi-solid droplets of this study does not qualify as a gas because it doesn't have a high vapor pressure at 50 °C and is not gaseous at 20 °C. It is not a liquid because its melting point is above 20 °C, consistent with our observation of their stability up to 95 °C (as shown in Fig. 3b, intact droplets in blue box). Therefore, according to ECHA's criteria of microplastic, the wax should be classified as a **solid**.

Microplastics in drinking water: There is no US Federal definition of microplastics. However, in 2020, the California Water Boards proposed the first specific definition of 'microplastics in drinking water' based on ECHA's criteria, explicitly including 'semi-solid' polymers within the definition of 'solid' substances [2, pp4]. They stated that 'solid' is defined as a substance or mixture which does not meet the definitions of a liquid or gas, thereby encompassing semi-solid polymers. They also highlighted that this classification is identical to the criterion in the definition of 'microplastics' proposed by ECHA.

We have also carefully verified that the sizes of the microparticles identified in our study align with the broad microplastic definitions proposed by other major regulatory bodies, including the U.S. National Oceanic and Atmospheric Administration and the U.S. Environmental Protection Agency. These agencies define microplastics as plastic pieces, particles or fibers smaller than 5 mm [3-4].

Based on these definitions and considerations, we believe it is appropriate to refer to the microparticles released into water in this study as microplastics.

We fully acknowledge that there are differences in physical and possibly chemical properties between the semi-solid droplets and the bulk polypropylene. As per your suggestion, we have added a section to the manuscript that highlights these differences. Additionally, we highlighted the need for further studies to understand the likely variations in attachment, translocation, accumulation and associated toxicity of waxy microparticles when they migrate into the environment or upon exposure to the human body. Finally, we have revised the manuscript title to highlight the unique properties of separated droplets and their relevance to microplastic release.

The revised title: Stress induced phase separation of **low molecular weight polymer** from plastics drives microplastic release into water.

We hope this clarification addresses your concerns. Thank you again for your valuable feedback, which has helped us improve the clarity and rigor of our manuscript.

Added Text to the Manuscript

It is noteworthy that the low MW and semi-solid nature of the released micro-sized particles investigated in this study is distinctly different from the more typically encountered rigid semicrystalline microplastics with higher molecular weights that are fragments of the parent plastic. Consequently, there are differences in their physical and possibly chemical properties compared to microplastics dominated by high molecular weight polymer that may lead to significant variations in the attachment, translocation, accumulation and associated toxicity whenever these particles enter the environment or come into contact with biological systems.

[1] European Chemicals Agency. ANNEX XV restriction report: proposal for a restriction-intentionally added microplastics. (Helsinki, Finland, 2019).

<https://echa.europa.eu/documents/10162/05bd96e3-b969-0a7c-c6d0-441182893720>

[2] California Water Boards. Proposed definition of 'microplastics in drinking water'. (2020). https://www.waterboards.ca.gov/drinking_water/certlic/drinkingwater/docs/dfntn_jun3.pdf

[3] U.S. National Oceanic and Atmospheric Administration. What are microplastics? (accessed by 27.11.2024). <https://oceanservice.noaa.gov/education/tutorial-coastal/marine-debris/md04.html#:~:text=Plastics%20can%20fragment%20into%20small,a%20pencil%20eraser%2C%20and%20smaller.>

[4] U.S. Environmental Protection Agency. Microplastics Research. (accessed by 27.11.2024). <https://www.epa.gov/water-research/microplastics-research>

Please see the revised text as it appears on Page 1 Lines 1-2, Page 8 Lines 314-321.

2. The experiment, schematically presented in Figure 4R, does not practically contribute or explain anything significant. Since the authors observe the migration of a molecular fraction with a mass of 22,000 g/mol, what more could have been expected in the case of a material with a molecular weight half that size?

In relation to the test presented in Fig. R5: In the original review, I postulated an increase in the degree of crystallinity of the polypropylene matrix, not a change in the crystallinity of the material within the droplets due to the annealing process at 95°C.

Response: We apologize for misunderstanding your initial suggestion and for adding unnecessary information to our response and manuscript. It is evident that standard amorphous polypropylene (PP) with a much lower molecular weight demonstrates higher mobility and diffusion capacity, making further experiments (Fig. R4) unnecessary. Additionally, the annealing experiment at 95 °C on standard amorphous PP is also not needed (Fig. R5).

As suggested, we have removed the relevant content in manuscript and Figure S1b and S9 in Supplementary Information. We will also contact the editor to remove Figures R4 and R5 from our last response. These revisions make the paper more concise and logical.

Please see the delete text as it appears on Page 4 Lines 132-138 in last manuscript and Figure S1b and S9 in last Supplementary Information.

3. Accordingly, the article may be accepted for publication if the authors remove the word "microplastic" from the title and content of the work.

Response: Many thanks for your nice suggestion. As suggested, we have revised the discussion section to highlight the unique properties of the waxy microparticles observed in this study and to suggest vital directions for future research. Additionally, we have revised the manuscript title to highlight the unique properties of separated droplets and their relevance to microplastic release. We sincerely appreciate your constructive feedback, which has significantly helped us enhance and refine our manuscript.

Added Content: It is noteworthy that the low MW and semi-solid nature of the released micro-sized particles investigated in this study is distinctly different from the more typically encountered rigid semicrystalline microplastics with higher molecular weights that are fragments of the parent plastic. Consequently, there are differences in their physical and possibly chemical properties compared to microplastics dominated by high molecular weight polymer that may lead to significant variations in

the attachment, translocation, accumulation and associated toxicity whenever these particles enter the environment or come into contact with biological systems.

Please see the revised text as it appears on Page 1 Lines 1-2, Page 8 Lines 314-321.

Dear Editor,

Thank you for your email on the status of our submission to Nature Communications.

We do not agree with the referee's argument and recommendation, for the following reasons:

1. The surface segregated material is amorphous and so does not have a melting point. Instead, the material progressively softens at higher temperatures. See details in Ref. [1]. We therefore cannot reasonably comply with the referee's request for clarification of the melting point process.
2. The material is not a "highly viscous liquid", as the referee claims. The schematic below shows that a viscous liquid flows under arbitrarily low levels of applied stress. We do not observe this pertinent feature in our experiments.

Instead, the material is observed to flow as a liquid only when the stress exceeds a critical yield stress. This is the well-known Bingham plastic type behaviour [2] detailed in Fig 2. At stresses below this critical value, the material is observed to behave as a solid and no droplets are observed at the surface (see schematic above and Figs 2c&d). To characterize the material as a "highly viscous liquid", as the referee suggests, is inconsistent with our reported measurements. Our measurements unequivocally indicate Bingham plastic type behaviour so that when the flow reaches the zero-stress surface the material becomes a solid Bingham plastic.

3. The released droplets are plastics, not pure polymer. As described on pp. 3 (line 114) they contain chemical additives – the other key ingredient of plastics – whose presence is demonstrated by the volume shrinkage of the droplets following exposure to solvents designed to dissolve the dominant additives in polypropylene.

Given that droplets are observed only when the critical stress is exceeded, the amorphous composition of the droplet material, its reduced molecular weight compared to the parent plastic, and that it is captured as globular entities in water (Fig 3C), we propose the following revised title:

Intrinsic stress induced phase separation of low molecular weight polymer from plastics drives the release of amorphous globular microplastics into water

We believe this title makes clear the distinction between this report and the rigid and irregularly shaped microplastic particles described in the literature, which are all based on polycrystalline polymers. The globular nature of the separated materials in water is emphasized in the revised text. All changes to the text are highlighted.

Finally, we re-iterate that we are introducing a new mechanism that results in the generation of a new type of "microplastic" (use of this term meets all regulatory criteria) [3-4]. To replace "microplastic" with "highly viscous liquid" would only generate confusion. The formation process we report is due to the intrinsic properties of the plastic itself, which is highly novel. This contrasts with microplastics whose generation is determined by extrinsic environmental factors. The revised text makes this point

even more clearly, and we do not think the readers of Nature Communications will be in any way confused.

We hope our paper is now acceptable for publication in its present revised form.

All the best,

Dunzhu, Liwen, Jing and John

References

[1] Giles Jr, H. F., Mount III, E. M. & Wagner Jr, J. R. Extrusion: the definitive processing guide and handbook. (William Andrew, 2004) pp520.

[2] Bingham, E. C. Fluidity and Plasticity. Vol. 2 439 (McGraw-Hill, 1922) pp216-217.

[3] European Chemicals Agency. ANNEX XV restriction report: proposal for a restriction-intentionally added microplastics. (Helsinki, Finland, 2019).

<https://echa.europa.eu/documents/10162/05bd96e3-b969-0a7c-c6d0-441182893720>

[4] California Water Boards. Proposed definition of 'microplastics in drinking water'. (2020).

https://www.waterboards.ca.gov/drinking_water/certlic/drinkingwater/docs/dfntn_jun3.pdf

Response to the referee

We would like to thank the editor and the reviewer for reviewing our manuscript. We agree with all the comments and the manuscript has been revised accordingly. All changes are highlighted in blue and can be found in the word document (main text-blue tracked.doc). Our responses to the specific comments of the reviewer are summarized below.

REVIEWER COMMENTS

Reviewer #4 (Remarks to the Author):

1, Intrinsic stress induced phase separation of low molecular weight polymer from plastics drives the release of amorphous globular microplastics into water

Dunzhu Li et al.

The manuscript is timely, interesting and provides novel results on the degradation of PP and PE.

Response: We sincerely thank you for your thoughtful review and kind comments. We greatly appreciate the time and effort you dedicated to assessing our manuscript. In response to your suggestions, we have carefully addressed each query and comment point-by-point and have made the necessary revisions to the manuscript.

2. At this point, I would like to mention that the manuscript focuses mainly on PP and PE is only mentioned once in the results section which seems to me an imbalance in the presentation of the results.

Response: We apologize for the lack of clarity regarding the presentation of polyethylene (PE) content in the manuscript. In the previous version, we summarized the key findings in the main text and provided comprehensive methods, experiments and results in the Supplementary Information to avoid excessively lengthening the main text.

In response to your suggestion, we have now incorporated more detailed data and analysis of PE into the main text. We have also clearly indicated the location of the PE-related content within the main text to benefit readers.

Relevant text in Supplementary Information: In addition to PP plastic sheets, we investigated stress-driven amorphous phase separation on the surfaces of standard semicrystalline polyethylene (PE) sheets (Fig. R1, Below). By configuring a PE sheet into a cantilever setup (Table S1), we observed that only the upper compressive side of the cantilever, approximately 3 mm from the clamped end, developed a significant number of droplets. These droplets typically ranged from 10-40 μm in lateral size and 0.3-2 μm in height after being exposed to a 95 °C oven (Fig. R1e-g). In contrast, no observable changes occurred on the tensile surface or the compressive side near the free end, consistent with observations from PP sheets. Raman spectroscopy analysis indicated the absence of the peak at 1416 cm^{-1} , typically associated with methylene bending¹. The degree of crystallinity in PE is also proportionally related to the intensity of this 1416 cm^{-1} peak (Fig. R1c)¹. The absence of this specific band confirmed that the migrated droplets from the PE sheet were also amorphous. Following the same experimental protocol as with the PP sheet, the normalized volumetric flow (\hat{v}) of amorphous PE (a-PE) droplets was obtained as 26.6 $\text{nm}^3/(\text{nm}^2\cdot\text{h})$ or nm/h , which is higher than that observed for PP under the same compressive stress level (around -5 MPa, Fig. 2c), showing that the specific nature of polymer material influences the migration speed of amorphous polymer. Additionally, PE sheets formed into a cylindrical shape were also immersed in 95 °C DI water for 4 hours. Post-exposure inspection revealed a high quantity of circular shapes formed by air bubbles on the surface of the PE

sheet (Fig. R1c). Residues of these partial circles were also observed, indicating dynamic processes of air bubble expansion and coalescence that substantially deformed these separated droplets. No a-PE droplets or circles were observed on the outer tensile surface of the cylindrical sheets. Although further research is required on other plastic types, the combined global market share of PP and PE, which accounts for 54%², underscores the vital importance of stress-driven phase separation in the plastics industry.

Revised text in main text: Compared to PP sheets, stress-driven flow onto the surface of semicrystalline polyethylene (PE) sheets is about 5.5 times more pronounced under the same temperature and compressive stress conditions (detailed in Supplementary Information, Fig. S14 and Text. S1). Air-bubble driven rearrangements of the droplets is also more pronounced in the case of PE and results in the formation of larger quantities of circles and circles residuals (Fig. 3b and Fig. S14c), indicating that the specific nature of polymer substantially influences the both the flow rate and the interaction of the amorphous polymer with water.

Please see the revised text as it appears on Page 6 Line 253, and Supplementary Information Page S3 and Page S18, Fig. S15.

References:

- [1] Strobl, G. & Hagedorn, W. Raman spectroscopic method for determining the crystallinity of polyethylene. *Journal of Polymer Science: Polymer Physics Edition* 16, 1181-1193 (1978).
- [2] Hahladakis, J. N., Velis, C. A., Weber, R., Iacovidou, E. & Purnell, P. An overview of chemical additives present in plastics: Migration, release, fate and environmental impact during their use, disposal and recycling. *Journal of hazardous materials* 344, 179-199 (2018).

Figure. R1 Analysis of surface stress driven a-PE phase separation and rearrangement in water environment. **a**, Optical image of virgin surface of standard PE sheet (low density). **b**, Optical image of PE compressive surface that close to clamped end after 95 °C oven heat. **c**, Optical image of inner compressed surface of PE sheet after 4hrs of 95 °C hot water exposure. **d**, Raman spectra of droplets from the surface of the stressed PE sheet, compared to the spectra from standard bulk PE sheet. **e-f**, AFM images of PE surface before and after 95 °C oven heat, both images have the same scale bars. **g**, The cross-section profiles of typical surface droplets in figure f.

3. term “microplastics”

The authors state that the “droplets are sticky gel-like substances”, which by definition are semi-solid substances. In the rebuttal, they also claim that, according to their interpretation of the current definitions, these droplets should be referred to as microplastics.

According to the EU 2023 (https://ec.europa.eu/commission/presscorner/detail/en/ip_23_4581) microplastics are defined as follows: The adopted restriction uses a broad definition of microplastics – it covers all synthetic polymer particles below five millimetres that are organic, insoluble and resist degradation. The adopted Annex XVII reads as follows (2023 Annex XVII to Regulation (EC) No 1907/2006 of the European Parliament and of the Council concerning the Registration, Evaluation, Authorisation and Restriction of Chemicals (REACH) as regards synthetic polymer microparticles): Synthetic polymer microparticles: polymers that are solid and which fulfil both of the following conditions: (a) are contained in particles and constitute at least 1 % by weight of those particles; or build a continuous surface coating on particles; (b) at least 1 % by weight of the particles referred to in point (a) fulfil either of the following

conditions: (i) all dimensions of the particles are equal to or less than 5 mm; (ii) the length of the particles is equal to or less than 15 mm and their length to diameter ratio is greater than 3.

The following polymers are excluded from this designation: (a) polymers that are the result of a polymerisation process that has taken place in nature, independently of the process through which they have been extracted, which are not chemically modified substances; (b) polymers that are degradable as proved in accordance with Appendix [X]; (c) polymers that have a solubility greater than 2 g/L as proved in accordance with Appendix [Y]; (d) polymers that do not contain carbon atoms in their chemical structure.

This roughly corresponds to the definition of microplastics by the International Organization for Standardization (ISO) ISO 24187:2023 (<https://www.iso.org/obp/ui/en/#iso:std:iso:24187:ed-1:v1:en>): 3.2 microplastic: any solid plastic particle insoluble in water with dimension between 1 µm and 1 000 µm (= 1 mm)

Note 1 to entry: Primary microplastics object represents a particle intentionally added to end-user products for example cosmetic means, coatings, paints etc. Secondary microplastics object can also result as a fragment of the respective item. Note 2 to entry: Microplastics have regular and irregular shapes (see ISO 9276-6:2008). Note 3 to entry: The defined dimension is related to the longest length of the particle.

[SOURCE:ISO/TR 21960:2020, 3.9, modified — Note 1 to entry was removed, all other Notes to entry were changed.]

In their response, the authors cited an older ECHA definition (2018) that still referred to semi-solid particles, which is no longer the case in the new definition. However, they further clarified how ECHA defines a solid: a substance or mixture that does not meet the definitions of a liquid or gas. This definition is still included in Annex XVII.

However, the analytical research and definition of MP is still an ongoing process and most studies on environmental MP pollution exclude waxes. One aspect that is also included in the definition of microplastics is the long-term persistence of the particles. No information is given here about the PP droplets, so this criterion can hardly be assessed. I assume that the droplets, due to their amorphous character, are likely to be rapidly degraded by microorganisms in the environment.

Given all these uncertainties, I would rather support reviewer3's opinion not to use the term microplastics, since PP droplets are semi-solid and not solid and have completely different properties (as pointed out by reviewer3 - lower MW, lack of crystallinity, and regioregularity of the chains) than "classic" microplastic particles. In almost all reports on environmental contamination, MPs are classified into fragments, foam, spheres or fibres. None of these properties fits gel-like PP-droplets.

I therefore recommend not to categorize the results found here under the general term "microplastics". I think the authors' findings are new and therefore this degradation product also deserves a new name.

I further recommend avoiding the term "microplastics" throughout the manuscript (in the title, abstract and main text body) to describe gel-like PP and PE droplets. Further, I strongly recommend not using the term "globular microplastics" (GMP) because "globular" merely means "spherical" and the droplets have nothing in common with spherical MP beads. Since gel-like PP and PE droplets appear to be a special form of plastic degradation products, which are not considered in the MNP research community, I would rather recommend discussing similarities, but especially differences, to "classic" microplastics, but only in the discussion section.

Response: We greatly appreciate your timely feedback regarding the evolving definitions from both regulatory bodies and the research community. We fully agree with your suggestion. Based on your input and considering the nature of the released material, we have replaced the term "microplastic" (MP) with "plastic micro-pollutant" (PMP). We have emphasized throughout the denatured state of the released material, in that it is altered in structure and composition from that of the parent plastic, and throughout the text we make the point that this material is different from traditional microplastics. As suggested, the term "microplastic" in the title, abstract, figures, main text and supplementary information has been replaced or removed.

We also removed the term "globular" through the paper to avoid any confusion, as it was not appropriate for the non-spherical droplets observed.

Furthermore, we have added a paragraph in the discussion section to compare our PMPs with traditional microplastics, focusing on potential differences in their behavior regarding attachment, translocation, accumulation and associated toxicity in environmental and biological systems. We

sincerely thank you for this insightful suggestion, which has substantially improved the clarity of this paper.

Please see the revised title, abstract on Page 1 (lines 24, 26, 35, 38 and 40), Figure 3e and 4b on Pages 21 and 22, and Supplementary Information Page S1 (title) and Page S18.

New added content on Page 6, Lines 241-242, Page 8 Lines 323-337; Other revisions in the main text are highlighted in blue.

Revised Title: Stress induced phase separation of low molecular weight polymers drives the release of denatured plastic micro-pollutants into water.

Added text 1 in discussion: It is important to note that the definition of microplastics is rapidly evolving both within the research community and regulatory bodies³⁻⁷. For instance, ECHA and California Water Boards initially included small-sized semi-solid plastics as microplastics^{4,7}. However, ECHA and EU later revised this definition, removing the reference to semi-solids^{5,6}. This highlights existing uncertainties in classifying amorphous and semi-solid PMPs as microplastics. Their denatured state, reduced molecular weight and amorphous structure compared to the original parent plastic, distinguish them from microplastics described in the literature⁸, which typically have higher molecular weights and are fragments of original polycrystalline plastics. Nonetheless our observations that plastic degradation under stress conditions results in both structure and compositional changes have strong parallels with degradation in the environment where reduced molecular weight and compositional changes have also been reported, albeit over much longer time scales⁹. Crucially, structure and compositional differences between PMPs and traditional MPs may lead to significant variations in the attachment, translocation, accumulation and associated toxicity whenever PMPs enter the environment or come into contact with biological systems.

Added text 2: Transfer of surface segregated droplets into the water generates plastic micro-pollutants (PMPs) that are denatured, with structure and composition different from that of traditional polycrystalline microplastics.

Added text 3: In contrast to the rigid and irregularly shaped MPs commonly found following environmental degradation, PMPs have a smooth appearance consistent with their amorphous structure (Fig 3c).

References:

- [3] Mitrano, D. M. & Wohlleben, W. Microplastic regulation should be more precise to incentivize both innovation and environmental safety. *Nature communications* 11, 1-12 (2020).
- [4] California Water Boards. Proposed definition of 'microplastics in drinking water'. (2020).
- [5] European Chemicals Agency. ANNEX XV restriction report: proposal for a restriction-intentionally added microplastics. (Helsinki, Finland, 2019).
- [6] European Commission. Amending Annex XVII to Regulation (EC) No 1907/2006 of the European Parliament and of the Council concerning the registration, evaluation, authorisation and restriction of chemicals (REACH) as regards synthetic polymer microparticles. (2023).
- [7] European Chemicals Agency. Note on substance identification and the potential scope of a restriction on uses of 'microplastics'. (2018)
- [8] Nava, V. et al. Plastic debris in lakes and reservoirs. *Nature* 619, 317-322 (2023).
- [9] Yang, J. et al. Effects of soil environmental factors and UV aging on Cu 2+ adsorption on microplastics. *Environmental Science and Pollution Research* 26, 23027-23036 (2019).

4. I have an additional concern regarding the methods used:

Methods: Microplastics release from stressed plastic sheet and from PP bottles

The authors write: PP sheets and stainless-steel O-rings were thoroughly rinsed by DI water (repeated 3 times at room temperature). Was the DI water filtered before use to avoid contamination since DI water usually contains MP? Which other QA/QC measures were taken to avoid contamination? Were any blanks taken at each sample processing step to test for contamination. This would be crucial to exclude that the measured particles and droplets originate from the samples and not from, for example, air contamination. Although the α -PP droplets can be distinguished from "classic" PP-MP, contamination of the sample with α -PP droplets cannot be excluded if the degradation mechanism shown here occurs frequently. Therefore, quality assurance/quality control measures must be reported, otherwise the reliability of the results cannot be fully assessed.

Response: Many thanks for the suggestion. Yes, the DI water used in our experiments was filtered through a 0.2 μm filter to prevent contamination. Specifically, the DI water was sourced from a Veolia UltraPure water system, which includes a Thermo Scientific™ Barnstead™ Nanopure unit equipped with a 0.2 μm absolute final filter. This system ensures the DI water meets stringent quality standards, with a conductivity of 1.5 $\mu\text{s/cm}$ and a resistivity of 18.2 M Ω , confirming its purity.

Additionally, we implemented several quality assurance and quality control (QA/QC) measures throughout the experiment to minimize potential contamination. These include:

(1) All equipment in contact with the samples was made from clean Borosilicate glass (3.3) and thoroughly cleaned before use.

(2) Particle-free nitrile gloves and 100% cotton-based laboratory coats were used. Gloves were washed with distilled water, followed by DI water, before each step.

(3) A control sample was tested for every five experimental samples to assess background PMP concentration in DI water and potential contamination during sample preparation. For the control sample, no plastic was placed in the beaker, and the same protocol as in Fig. 3a was followed (e.g., filling a glass beaker with 95°C DI water, soaking for 4 hours, cooling, filtering the water sample, followed by ethanol filtration to remove plastic additives, and detecting using Raman and SEM). No PP PMPs were detected in the control samples, confirming the reliability of our method.

These QA/QC measures were crucial for ensuring the accuracy and reliability of the results. As suggested, we provided all above information in the main text. The control sample results were also incorporated in figure 3e and 4b.

Please see the revised text as it appears on Page 12 Lines 500-516, and Figure 3e and 4b on Pages 21 and 22.

5. minor aspects: "Water environment" is unusual wording – it is either "in contact with water" or "aquatic environment"

Response: We have carefully reviewed and replaced "water environment" with "aquatic environment," as suggested.

Please see the revised text as it appears on Page 8 Line 314.

6. “suggesting that direct release of MPs induced by aquatic factors such as air bubbles is likely to occur in parallel with extrinsic environmental factors such as UV-induced degradation or mechanical abrasion.” Aquatic factor is not precise. Environmental factors can either be biotic or abiotic, and even abiotic factors, which are probably addressed here, are quite diverse and I think that only physical factors such as turbulence may lead to the release.

Response: Many thanks for the suggestion. We agree that "physical factors" is more precise. We have revised the sentence to better describe the potential release mechanisms.

Revised sentence: “suggesting that their release as plastic micro-pollutants (PMPs) into water could be influenced by *physical factors* such as air bubbles and *turbulence* in parallel with extrinsic environmental factors such as UV-induced degradation or mechanical abrasion.”

Please see the revised text as it appears on Page 6 Lines 221-222.

7. *Although the exact quantity of nanoplastics was not quantified*

I would recommend using “exact number” instead of quantity.

Response: As suggested, we have replaced "exact quantity" with "exact number" to improve clarity.

Please see the revised text as it appears on Page 7 Line 267.

Response to the referee

We would like to thank the editor and the reviewer for reviewing our manuscript. We agree with all the comments and the manuscript has been revised accordingly. All changes are highlighted in blue and can be found in the word document. Our responses to the specific comments of the reviewer are summarized below.

REVIEWER COMMENTS

Reviewer #4 (Remarks to the Author):

The term plastic micro-pollutants (PMPs) is not very common, but it is used as a synonym for “classical” microplastics. The authors also use the term in this sense (“Micro-pollutants released from semicrystalline plastics are known to threaten the environment and human health”), so they do not draw any conceptual distinction between microplastics and their newly discovered amorphous polymer droplets. I would therefore strongly advise to distinguish between “classical” microplastics and amorphous polymer droplets (APD) in the manuscript and therefore to use the term APD in the title and in the manuscript.

Response: Thank you for taking the time and effort to review our manuscript once again. We fully agree that highlighting the amorphous polymer aspect is essential to this research. Following your suggestion, we have replaced the term “denatured plastic micro-pollutant” with “amorphous polymer micro-pollutant” to underscore the amorphous nature of the released material.

Regarding the term “droplet,” we refrained from using it for the following reasons:

1. The term “droplet” typically implies a spherical or near-spherical shape, which might be misconstrued as being similar to polymer microspheres or microbeads — a confusion we want to avoid, and I believe we both share this concern.
2. Although the amorphous polymer appears “droplet-like” on the plastic surface, the material has a non-zero-yield stress and its shape becomes deformed and rearranged following release and capture by membranes (see Figure R1 and Figure 3c).
3. “Micro-pollutant” is an umbrella term that applies to small-sized contaminants released from any material including plastics that may harm human health and the environment. The term accurately conveys the size of the released material in our case. Classical microplastics are one example of a plastic-derived micro-pollutant, stress-driven amorphous polymer release is another.
4. Crucially, to avoid any possible confusion we make the clear throughout the manuscript the differences the released amorphous polymer and classical MPs.

We deeply appreciate your insightful comments, which have significantly improved the clarity of our manuscript. We have updated the terminology throughout the text and figures accordingly. Thank you again for your invaluable feedback.

Revised title: Stress induced phase separation in plastics drives the release of amorphous polymer micro-pollutants into water

Please see the revised title, abstract on Page 1 (lines 31, and 32), Figure 3e and 4b and relevant revisions throughout the paper.

Figure. R1 SEM image of typical amorphous polymer materials released into water and captured on membrane filter surface (red circles).